Subject Areas:
psychology

Keywords:
memory, episodic memory, memory integration

Authors for correspondence:
Emma James
e-mail: emma.james@york.ac.uk
Aidan J. Horner
e-mail: aidan.horner@york.ac.uk

# Make or break it: boundary conditions for integrating multiple elements in episodic memory

Emma James[1], Gabrielle Ong[1], Lisa M. Henderson[1,2] and Aidan J. Horner[1,2]

[1]Department of Psychology, and [2]York Biomedical Research Institute, University of York, York, UK

EJ, 0000-0002-5214-0035; AJH, 0000-0003-0882-9756

Event memories are characterized by the holistic retrieval of their constituent elements. Studies show that memory for individual event elements (e.g. person, object and location) are statistically related to each other, and that the same associative memory structure can be formed by learning all pairwise associations across separated encoding contexts (person–object, person–location, object–location). Counter to previous studies that have shown no differences in holistic retrieval between simultaneously and separately encoded event elements, adults did not show evidence of holistic retrieval from separately encoded event elements when using a similar paradigm adapted for children (Experiment 1). We conducted a further five online experiments to explore the conditions under which holistic retrieval emerges following separated encoding of within-event associations, testing for influences of trial length (Experiment 2), the number of events learned (Experiment 3a) and stimulus presentation format (Experiments 3b, 4a, 4b). Presentation of written words was optimal for integrating elements across encoding trials, whereas the addition of spoken words disrupted integration across separately presented associations. The use of picture stimuli also produced effect sizes smaller than those of previously published research. We discuss the ways in which memory integration processes may be disrupted by these differences in presentation format. The findings have practical implications for the utility of this paradigm across research and learning contexts.

# 1. Introduction

When we remember an event, we probably retrieve many different types of information from the experience—such as where we are and the people and items present [1]. A characteristic feature of episodic memory is that we are more likely to retrieve these event elements together than we are to retrieve each element in isolation [2]. Recent studies demonstrate that this holistic retrieval of event elements results from the associative structure of these memories, composed of links between each element present [3]. These associations enable all event elements to be retrieved in the presence of a partial cue, via processes of *pattern completion* [4–6]. Interestingly, holistic retrieval can also emerge following the spaced encoding of each association separately, despite not being experienced at the same time [3]. Thus, mnemonic representations that support holistic retrieval can be formed by integrating overlapping information across separate encoding trials, in the absence of a clear spatio-temporal context. Here, we present six experiments that test the boundary conditions of this separated encoding procedure in forming episodic-like memories. In doing so, we aimed to better understand the extent to which this paradigm can be adapted to examine related memory processes, and the processes involved in binding together elements in episodic memory. Mnemonic integration is a fundamental process that allows us to generalize across experiences and infer new relationships between elements not directly associated. It is, therefore, crucial that we understand the experimental conditions that do, and do not, promote integration.

Experimental evidence for holistic retrieval in episodic memory has been provided by studies that assess the relationship between memory for different aspects of the event (e.g. [7–12]). In one paradigm developed by Horner & Burgess [2], participants encode a series of multi-element 'events', formed from a person, object and location. For example, participants might be presented with *Barack Obama–supermarket–pencil case* as written words, and be asked to visualize the three elements interacting as vividly as possible. This visualization process is designed to facilitate the integration of the elements into an event-like memory. At retrieval, participants are tested on their memory for each of the pairwise associations independently (*Obama–supermarket; Obama–pencil case; supermarket–pencil case*). If pattern completion permits the holistic retrieval of all event elements from a partial cue, then memory for within-event pairs should be related to each other: retrieving the correct location *supermarket* when cued with *Obama* should also make it more likely that one retrieves *pencil case* from the same cue. Evidence for this retrieval dependency has been demonstrated in a number of studies using this paradigm [2,3,13], and can be seen in children as young as 4 years old [14].

One possibility is that within-event similarity in memory performance may result from fluctuations in attention at encoding, rather than pattern completion processes at retrieval. To distinguish between processes that might influence dependency at encoding versus retrieval, Horner & Burgess [3] conducted a *separated* encoding condition in which each of the three pairwise associations were presented separately throughout the encoding phase (e.g. *Obama–supermarket; Obama–pencil case; supermarket–pencil case* learnt across three separate encoding trials separated by several minutes and encoding trials of other elements). Although not presented at the same time, creating a similar associative structure between event elements also resulted in retrieval dependency. Dependency following *separated* encoding was statistically indistinguishable from dependency following *simultaneous* presentation of event elements, providing that all within-event associations were learned. That is, this retrieval dependency cannot be attributed to fluctuations in encoding strength, but the complete and coherent associative structure of the event memory [3]. In this sense, the processes studied here probably differ from a related area of the literature that examines inference for *non*-encoded information across overlapping memories (e.g. [15,16]). Although participants are typically able to draw inferences from overlapping information (e.g. inferring the relationship between A–C after the encoding of A–B and B–C), this does not necessarily mean that integration has occurred (as inference may occur via associative retrieval at the point of inference, as opposed to the full integration of information at encoding). Indeed, previous research has shown that retrieval dependency is not seen for A–B, B–C overlapping pairs, despite the ability to infer A–C, suggesting that integration may not be driving inference in this paradigm [3]. Evidence of pattern completion at retrieval following the encoding of all within-event associations has since been supported by neuroimaging studies, which show that hippocampal activity at retrieval is associated with element-related neocortical activity—even for those event elements not directly tested [17,18]. Furthermore, Joensen *et al.* [19] demonstrated that this retrieval dependency is consistent over time, such that events are probably forgotten in an all-or-none manner. Thus, integration from separated encoding trials

provides a useful paradigm for studying the structure of memory representations without the confound of attention during encoding.

However, while previous studies have not found a difference in dependency between simultaneous and separately encoded event elements when using this paradigm, it seems likely that such differences exist: we do not typically retrieve all common event elements experienced across different episodes. Indeed, the above neuroimaging studies also identified additional neural activity upon the presentation of the third association during encoding [18]; anterior hippocampal activity on the third (final) encoding trial predicted subsequent memory for the within-event associations encoded on the first and second encoding trials. This evidence highlights that an additional process might be at play for the separated (versus simultaneous) encoding condition—integrating the three associated elements into a coherent representation on the final trial. While this separated encoding paradigm has proven a useful research tool, understanding how the resulting memories may differ from event representations encoded in the same temporal context is vital for understanding its limitations. Better understanding the processes at encoding and how different factors might influence subsequent retrieval dependency is important for three key reasons. First and foremost, understanding the limitations of this paradigm is of practical use to researchers designing related studies of episodic memory. Knowing the conditions under which retrieval dependency is established will avoid experimental designs that fail to capture holistic retrieval for reasons outside those of theoretical interest. Second, the conditions under which associated elements form episodic-like representations are of theoretical relevance for understanding how episodic memories are formed and segmented. Third, it may have applied benefits in understanding whether the paradigm has use outside of research contexts. The creation of such robust memory structures that are less prone to decay [19] *could* benefit learning in educational settings, but would require significant adaptation to be relevant to learning real-world information. Understanding and documenting the limitations of this paradigm is essential for progress in all of these domains.

Here, we present six experiments that examine the conditions under which retrieval dependency from separately encoded event elements emerges. In Experiment 1, we adapted Horner & Burgess's [3] paradigm to make it appropriate for a developmental sample, yet failed to find evidence of retrieval dependency in a well-powered adult sample. We present a further five online experiments to explore the potential impact of each adaptation on disrupting retrieval dependency in this paradigm, including trial timings (Experiment 2), the quantity of information learned (Experiment 3a) and the modality of the stimuli (Experiments 3a, 4a, 4b; table 1). In doing so, we hoped to better understand the conditions under which retrieval dependency can occur from separately encoded associations. To summarize, the modality of the stimuli presented appears to play a key role in whether retrieval dependency is seen when associative structures are built up across three separate encoding trials.

# 2. Experiment 1 methods

Experiment 1 constitutes an exploratory analysis of an adult dataset collected as part of a larger developmental study (http://osf.io/br23e) comparing dependency in both simultaneous and separated encoding conditions. All experiments presented were approved by the Psychology Research Ethics Committee at the University of York.

## 2.1. Participants

Forty-five adults aged 18–30 years were recruited from the York Psychology Participant Pool (42 female; mean age = 21.3 years). All were required to be at least highly fluent English speakers. An additional 10 participants completed the tasks but were excluded on the basis of near ceiling levels of performance (greater than or equal to 95% across both conditions; described below). The session lasted approximately 35 min in total, and participants received either course credit or £5 payment.

## 2.2. Stimuli

Each 'event' comprised three elements: an animal, object and location. Animal characters were used instead of famous people (e.g. [2]) to make the task accessible to a wide range of age groups. Two sets of items were developed that were matched on age of acquisition rating [20], concreteness [21] and number of syllables. Within each set, items of each element type were pseudorandomly combined to create fixed events (e.g. *cow-backpack-post office*), avoiding strong pre-existing semantic associations (e.g. *book-library*).

**Table 1.** Summary of experimental design differences and dependency effects across all experiments.

| experiment: manipulation | experimental set-up | | | presentation modality | | | results | |
|---|---|---|---|---|---|---|---|---|
| | sample $n$ | events $n$ | trial time (s) | images | spoken words | written words | effect size ($d$) | $BF_{01}$ |
| Exp. 1: simultaneous encoding[b] | 45 | 15 | 4 | Y | Y | — | 0.70[a] | 0.001 |
| Exp. 1: separated encoding[b] | 45 | 15 | 3 | Y | Y | — | −0.06 | 5.80 |
| Exp. 2: constrained timings[b] | 45 | 15 | 3 | Y | Y | — | 0.28 | 1.23 |
| Exp. 2: original timings[b] | 45 | 15 | 6 | Y | Y | — | 0.21 | 2.47 |
| Exp. 3a: all 30 items | 45 | 30 | 6 | Y | Y | — | 0.25 | 1.76 |
| Exp. 3b: written words | 20 | 30 | 6 | — | — | Y | 0.73[a] | 0.09 |
| Exp. 4a: pictures only | 20 | 30 | 6 | Y | — | — | 0.45 | 0.83 |
| Exp. 4a: words | 20 | 30 | 6 | — | Y | Y | 0.07 | 4.13 |
| Exp. 4b: pictures only | 20 | 30 | 6 | Y | — | — | 0.39 | 0.82 |
| Exp. 4b: words | 20 | 30 | 6 | — | Y | Y | 0.00 | 4.30 |

[a]Retrieval dependency that is significantly greater than would be predicted by the independent model.
[b]Conditions administered in a within-subjects design; all other experimental manipulations were between-subjects.

While previous studies have presented the stimuli using written words, we used spoken words (recorded by a female native English speaker) to minimize demands on reading ability in the developmental study. To reduce demands on working memory and make the task more engaging, we additionally sourced cartoon illustrations for each item using a Web-based image search.

## 2.3. Design and procedure

Participants were asked to help a fictional character, *Agent Arnie*, to remember things he sees on his adventures. Each participant completed the activities in two different encoding conditions. In the simultaneous encoding condition, all three event elements (*animal–object–location*) were presented at the same time. In the separated encoding condition, each pairwise association between the event elements were presented separately (*animal–object; animal–location; object–location*). Participants completed the encoding and retrieval tasks for one condition before proceeding with the second condition. The order of encoding conditions and the stimulus list assigned to each were counterbalanced across participants (with no effect of condition order on retrieval dependency, $p > 0.90$). The experimental tasks were programmed using OpenSesame ([22]; scripts available at http://osf.io/vqzh8). Participants also completed a matrix reasoning task after finishing both experimental conditions (for purposes of comparing developmental samples; data not presented but available online).

### 2.3.1. Encoding

During the encoding task, participants were presented with two (separated encoding condition) or three (simultaneous encoding condition) item illustrations, and heard each one named aloud through headphones. They were instructed to try and remember each set of items for a subsequent memory

test, and that they should visualize the items interacting to help them. The images remained on screen for an additional second after they had been named, totalling a trial time of 3 s (separated encoding condition) or 4 s (simultaneous encoding condition). The next set followed after a 500 ms interval. Note that these encoding trials are shorter than those used in previous studies using this paradigm (minimum 6 s), designed to avoid adults performing at ceiling levels in an experiment otherwise simplified for children. The total encoding time per event was 4 s for the simultaneous encoding condition and 9 s for the separated encoding condition (i.e. three pairwise associations encoded for 3 s each)—a difference that mirrors previous studies using this design.

For the simultaneous encoding condition, the events were presented in a randomized order in a single block of trials. For the separated encoding condition, one pairwise association from each event appeared within each of three blocks, allowing a short break in between blocks. The association type presented first was balanced across events (and counterbalanced across participants), such that within any block, there were five of each animal–object, animal–location and object–location pair types.

### 2.3.2. Retrieval

There were six retrieval trials per event, testing each of the three pairwise associations in both directions (i.e. cue *animal* retrieve *object*; cue *object* retrieve *animal*). Each participant was tested on every pairwise association once before completing the second set of tests in the opposite direction, the order of which was counterbalanced across participants. Within each of the test sets, there were three blocks of trials that each contained five of each association type (totalling 15 trials, one per event). The order of the three blocks was again counterbalanced across participants using a reduced latin square design.

For each retrieval trial, the cue picture was presented at the top of the screen, and the corresponding spoken word played through the headphones. Participants were asked to choose which of four numbered pictures underneath was seen with the cue picture, and select their answer using a key press response. Trials timed out after 3 s, and missing responses ($M = 0.04$, s.d. $= 0.04$) were counted as incorrect. Again, this trial time is reduced relative to previous studies (6 s) to avoid issues of high performance. Participants were given optional breaks after every other block.

## 2.4. Analyses

We excluded and replaced participants who averaged greater than or equal to 95% across both encoding conditions ($n = 10$). Analyses were conducted in R [23] using packages *lme4* [24] and *lmerTest* [25]. The data and analysis scripts for all experiments can be accessed at http://osf.io/cqm7v.

### 2.4.1. Accuracy

To assess differences in accuracy across encoding conditions, we fitted a generalized linear mixed effects model with encoding condition as a predictor. Encoding condition was effect coded such that a positive $\beta$-value would reflect higher accuracy in the simultaneous encoding condition. We started with random intercepts for both participants and events (i.e. the fixed triplets of elements), and tested whether additional intercepts (association type or pair type) and random slopes for the effect of encoding condition improved model fit under a liberal threshold ($p < 0.2$; Barr *et al.* [26]). The final model included random intercepts and slopes for both participant and event variability; and random intercepts for the pair type tested.

### 2.4.2. Dependency

We used individual retrieval trial accuracy to compute retrieval dependency: a measure that indicates whether retrieval of an association is statistically related to the retrieval of the other associations from the same event. This statistical dependency was computed as in previous studies (e.g. [2]), providing a measure of dependency in each participant that is scaled for their overall accuracy. To compute dependency in a participant's data, six contingency tables were formed—one for each pair of associations that shared a common cue or common retrieval target. For example, one table documents performance on tests that cue using the animal, computing contingencies in performance between pairs that require retrieval of the object and those that require retrieval of the location. The test pairs from all 15 events per condition are entered into the contingency table according to whether each pair was correct or incorrect, and the proportion of joint retrieval (and non-retrieval) is computed by

calculating the proportion of events that showed contingent accuracy from the common cue/target (correct–correct, incorrect–incorrect). For each participant, this proportion of joint retrieval is averaged across all six tables, producing a final measure in the data.

Given that this proportion of joint retrieval measure is influenced by accuracy (i.e. a participant with very high or very low accuracy would show many contingent responses), it is compared with a model that predicts the value if there was no relationship in memory for the pairs. Contingency tables for this *independent model* are formed by multiplying the independent probabilities for successful (or unsuccessful) retrieval for each retrieval pair. For example, the proportion of joint retrieval is calculated as $(P_{AB} \times P_{AC}) + ((1 - P_{AB}) \times (1 - P_{AC}))$, for the contingency table for the joint retrieval of animal (B) and object (C) when cued by location (A). The difference between the proportion of joint retrieval in the data and the independent model ([proportion of joint retrieval(data)] – [proportion of joint retrieval(indModel)]) formed the 'dependency' variable used for analysis, such that a positive value provides evidence for retrieval dependency (taking into account overall accuracy). If performance is very high, however, then the proportion of joint retrieval in the data cannot be higher than in the independent model, and we exclude participants at ceiling performance (greater than or equal to 95%) to avoid this issue. Note that this measure of retrieval dependency is the same measure that has been used in studies demonstrating hippocampal CA3 involvement in holistic recollection [17] and provides a robust measure of event memory over time [19].

In this exploratory analysis, we compare dependency across conditions using a linear mixed model, before testing for evidence for dependency in each condition using one-sample tests against 0. For the latter, we also report Bayes factors describing evidence in favour of the null hypotheses (computed using the *BayesFactor* package, [27]). This means that a Bayes factor greater than 1 favours the null hypothesis over the alternative hypothesis that dependency is present, with greater than 3 considered moderate evidence for the null and less than 0.33 as moderate evidence for the alternative hypothesis (see [28]). For all Bayesian analyses, the prior was a Cauchy distribution with the default $r = 0.707$, centred at 0 (i.e. to test the hypothesis that there was no retrieval dependency). We conducted robustness checks at different Cauchy widths to ensure that our conclusions were not unduly influenced by our choice of prior (presented in the OSF output files, or at https://osf.io/j5fpu).

# 3. Experiment 1 results

Trial- and participant-level data can be accessed for all experiments via the Open Science Framework (https://osf.io/cqm7v). Descriptive statistics are presented in figures 1 and 2—made using *ggplot2* [29] and *ggpirate* [30]—and a table of all summary statistics is available online (https://osf.io/yscxp).

## 3.1. Accuracy

Retrieval performance was well above chance in both the simultaneous ($M = 0.71$; s.d. $= 0.19$) and separated ($M = 0.73$; s.d. $= 0.22$) encoding conditions (figure 1a). There was no significant difference in accuracy between the two conditions ($p = 0.214$).

## 3.2. Dependency

Counter to our original hypotheses and to published data [3], there was a difference in dependency across the two conditions ($\beta = 0.04$, s.e. $= 0.01$; $t = 4.63$; $p < 0.001$): dependency was greater in the simultaneous encoding condition ($M = 0.04$, s.d. $= 0.06$) than the separated encoding condition ($M = 0.00$; s.d. $= 0.04$). We further explored this difference by conducting one-sample $t$-tests for each condition to test for evidence of dependency greater than 0. There was evidence for dependency in the simultaneous condition ($t_{44} = 4.67$, $p < 0.001$, $d = 0.70$; $BF_{01} = 0.001$), but no evidence for dependency was observed in the separated condition ($t_{44} = -0.37$, $p = 0.712$, $d = -0.06$; $BF_{01} = 5.80$), with moderate evidence in favour of the null hypothesis (figure 1b).

# 4. Experiment 1 discussion

Experiment 1 showed a clear difference in dependency between event elements encoded simultaneously and those experienced across separate encoding trials. This finding contrasts with the results of previous studies demonstrating no differences in dependency between simultaneous and separated encoding

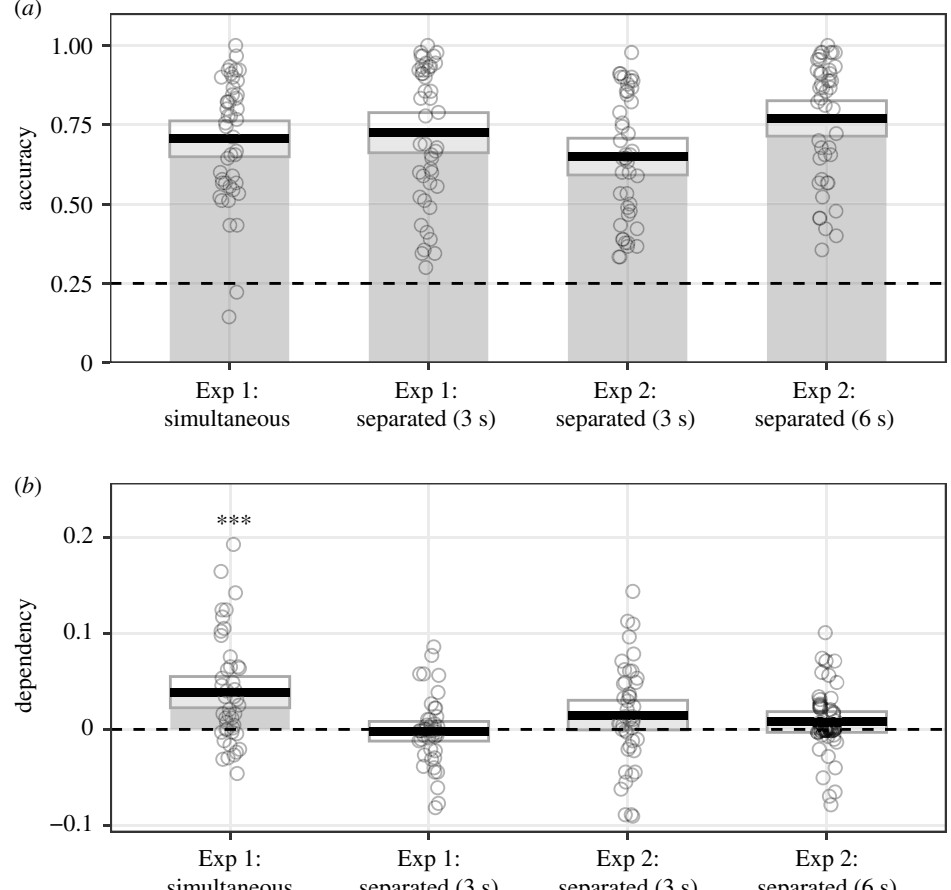

**Figure 1.** Mean participant scores in Experiments 1 and 2, plotted by encoding type (simultaneous versus separated) and trial timing condition (3 versus 6 s); for (a) proportion correct (dashed line represents chance level performance) and (b) dependency (dashed line represents no more dependency in the data than would be predicted by the independent model; ***significant dependency above 0 at $p < 0.001$). Black horizontal lines represent the mean, and surrounding boxes 95% confidence intervals. Each circle marks average performance of a single participant.

conditions [3,13]. We found clear evidence for dependency in the simultaneous encoding condition, where the event elements were retrieved holistically. This large effect size ($d = 0.70$) is comparable to those found in previous studies (range 0.5–1.26). However, there was very little evidence of holistic retrieval when each pairwise association was presented separately at encoding. The data in this condition showed reasonable evidence in favour of a null effect, despite dependency under separated encoding conditions being highly replicable in previous studies.

A key difference between this experiment and previous studies was that we decreased trial timings at both encoding and retrieval. While previous studies have successfully shown memory dependency from separate encoding when participants were given 6 s to encode and retrieve each trial (e.g. [19]), we allowed only 3 s per encoding and retrieval trial. This difference was implemented to prevent adults performing at ceiling levels in a task otherwise designed to be administered with children, but may have interfered with holistic retrieval in two ways. First, the reduced trial times at encoding may have been too short to allow participants to integrate each trial with previous event trials. Second, reduced trial times at retrieval may have limited the opportunities for pattern completion processes to support retrieval on any given trial. Before testing these possibilities in turn, we first sought to bring back dependency by increasing trial timings to those of original studies.

## 5. Experiment 2 methods

In Experiment 2, we set out to replicate the lack of dependency for the separated encoding condition with encoding and retrieval time constrained to 3 s. Importantly, we tested the pre-registered hypothesis that retrieval dependency would re-emerge in a condition where encoding and retrieval trial times were

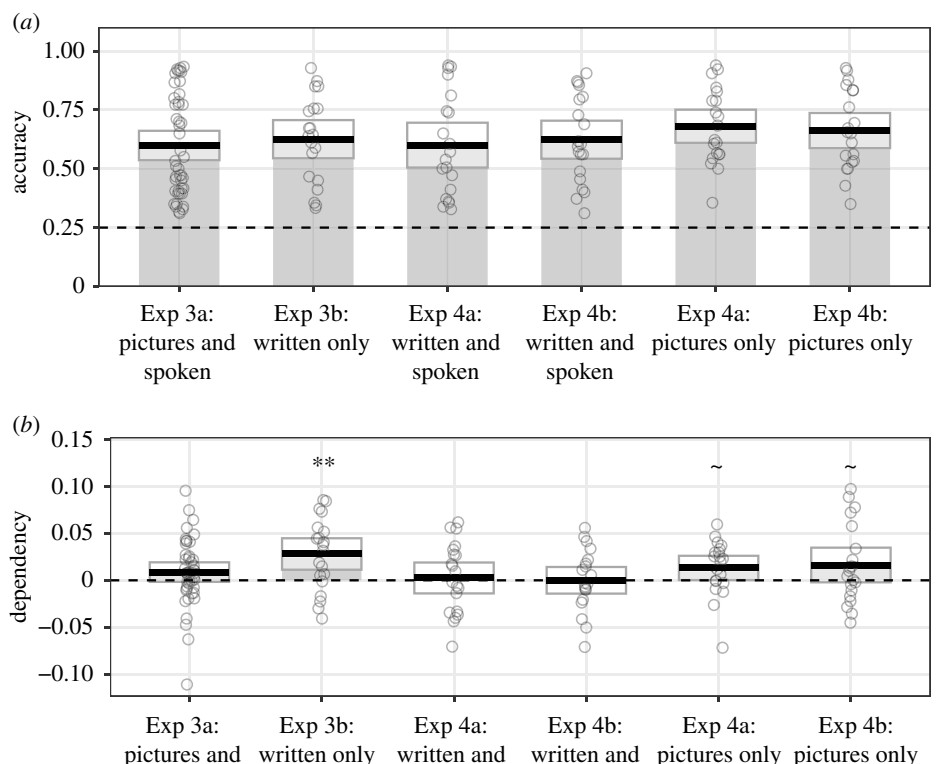

**Figure 2.** Mean participant scores in Experiments 3 and 4, plotted by stimulus modality; for (*a*) proportion correct (dashed line represents chance level performance) and (*b*) dependency (dashed line represents no more dependency in the data than would be predicted by the independent model; **significant dependency above 0 at $p < 0.01$, ~samples that show significant dependency when combined in exploratory analyses). Black horizontal lines represent the mean, and surrounding boxes 95% confidence intervals. Each circle marks the average performance of a single participant.

increased to 6 s (https://osf.io/ht9aq). While Experiment 1 was conducted in person, Experiments 2–4 were conducted online.

## 5.1. Participants

We invited 18-to-35-year-old native English speakers via Prolific (www.prolific.co) to take part in the study. The 45 participants eligible for inclusion in the analysis had a mean age of 27.1 years (range 19–34); with 24 reporting as female, 20 male and one other. Additional participants were excluded according to pre-registered criteria: discontinued from the study after failing one/more attention trials during encoding ($n = 12$; detailed below); failing to meet the specified age criteria ($n = 1$), performing at floor (less than or equal to 30%; $n = 5$) or at ceiling (greater than or equal to 95%, $n = 9$). Participants were paid £4 upon completion of the study.

## 5.2. Design and procedure

All experimental tasks were re-programmed using Gorilla [31], and can be accessed online (https://gorilla.sc/openmaterials/59473). Participants completed the experiment in a single sitting, lasting approximately 25–30 min. Trial timing was a within-subjects manipulation, and so each participant completed a *constrained* timing condition (encoding and retrieval trials 3 s each, as above) and an *original* timing condition (encoding and retrieval trials 6 s each, as in previous studies). Participants completed the encoding and test for each of the two trial timing conditions separately, with the order counterbalanced across participants. Prior to beginning the experiment, participants passed an audio screening check to ensure that their browser enabled automatic playing of sound files.

### 5.2.1. Encoding

The encoding task was identical to the separated encoding condition in Experiment 1, with two exceptions. First, we manipulated trial timings across two conditions: the constrained trial timings

were identical to Experiment 1 (leaving each pair on screen for 3 s), whereas the original trial timings left the pairs on screen for 6 s. Second, we incorporated attention trials to screen out participants who were not paying attention to the task, given the passive nature of the encoding task and reduced experimenter control during online testing. To avoid incorporating a secondary task, the trials simply required participants to press the spacebar on their keyboard within a 3 s window. One trial was included per block, and was randomized alongside the encoding trials. Participants who did not pass all three attention trials were discontinued from the study. We also asked participants to describe their strategy at the end of the study to check for alternative memory aids, but no participants were excluded on this basis.

### 5.2.2. Retrieval

The retrieval task was identical to Experiment 1, but also varied with the trial timing manipulation. The constrained trial timings were identical to Experiment 1 (timeout after 3 s), whereas the original timing condition gave participants up to 6 s to respond. Missing responses in either condition were counted as incorrect (constrained: $M = 0.03$, s.d. $= 0.03$; original: $M = 0.01$, s.d. $= 0.01$).

## 5.3. Analyses

Relative to Experiment 1, we pre-registered an additional exclusion threshold for near-chance performance (less than or equal to 30%), to avoid including participants who were not properly engaging with the task. We tested whether accuracy and dependency differed between the constrained and original trial timing conditions. Our approach was identical to Experiment 1 for accuracy, using timing condition rather than encoding type as the fixed effect of interest. The final model for accuracy incorporated random slopes and intercepts for participants, and intercepts only for the events and association types. For dependency, we first tested for the presence of dependency in each condition (constrained and original timing), and additionally provide (non-pre-registered) Bayes factors to aid in interpretation. We then modelled the differences between the two timing conditions; only participant intercepts could be included in the dependency model.

# 6. Experiment 2 results

## 6.1. Accuracy

Retrieval performance in the constrained condition was slightly lower than in Experiment 1 (probably due to the sample differences associated with online testing), but still well above chance ($M = 0.65$, s.d. $= 0.20$; figure 1$a$). As would be expected, retrieval performance was significantly higher in the original timing condition ($M = 0.77$, s.d. $= 0.19$), which allowed more time to both encode and retrieve each pair. This difference between conditions was statistically significant ($\beta = 0.83$, s.e. $= 0.16$; $Z = 5.19$; $p < 0.001$).

## 6.2. Dependency

In contrast with our hypothesis that there would be significant dependency in retrieving elements using the original trial timings, dependency remained very low in this condition ($M = 0.01$, s.d. $= 0.04$; $t_{44} = 1.41$, $p = 0.167$, $d = 0.21$; $BF_{01} = 2.47$), with evidence in favour of the null (figure 1$b$). There was slightly greater evidence of dependency in the constrained condition ($M = 0.01$, s.d. $= 0.05$) that was not statistically significant against 0 ($t_{44} = 1.89$, $p = 0.066$, $d = 0.28$; $BF_{01} = 1.23$). Dependency was not significantly different between the two timing conditions ($\beta = -0.01$, s.e. $= 0.01$; $t = -0.75$; $p = 0.455$).

# 7. Experiment 2 discussion

Experiment 2 again failed to find evidence for retrieval dependency in the condition with constrained trial timings but, counter to our hypotheses, neither was retrieval dependency present when returning to 6 s trial timings. While we note increased dependency for the 3 s condition in Experiment 2 relative to the identical condition in Experiment 1, this was not statistically significant and the Bayes factor did not favour the experimental hypothesis over a null effect. As such, it is not appropriate to

interpret this further. Although neither condition showed strong evidence in favour of a null effect, our large sample size had statistical power greater than 0.99 to detect the average published effect size ($d = 0.86$) using this paradigm, and 0.90 for the smallest published effect size. This suggests that the current adaptations may at least reduce dependency relative to previous studies, leaving us under-powered to detect such effects here.

Given that trial timings did not seem to be the issue, we did not continue with the second pre-registered experiment aimed to test whether encoding or retrieval times were most influential.

# 8. Experiment 3a methods

In Experiments 3a and 3b, we sought to bring the present experiments closer in line with previous studies: first, by increasing the number of events encoded, and second by returning to written word presentation format. While we had simulated varying numbers of events from previous data [19] to establish that 15 events should be sufficiently powered to detect evidence of dependency, this does not account for the different demands on memory during the encoding process. Similarly, although previous studies have successfully shown evidence of dependency with 18 events per condition (e.g. [3]), these were interleaved with other conditions during learning (rather than the encode–test–encode–test structure used in Experiments 1 and 2). Perhaps then, dependency may be more likely to emerge when one is forced to prioritize some elements of new learning over others. To test the hypothesis that the reduced number of events might be responsible for reducing dependency in the present experiments, Experiment 3a incorporated a single study-test session with all 30 events (https://osf.io/v7n4y).

## 8.1. Participants

We included 45 eligible participants recruited via Prolific as in Experiment 2, with the additional criterion that participants must not have taken part in the earlier experiments. Fifteen further participants were discontinued after failing one/more attention trials during encoding; two participants were excluded for floor performance and 14 for ceiling performance. The final sample had a mean age of 27.36 years (range 19–35), with 35 reporting as female, nine male and one other.

## 8.2. Design and procedure

The task was identical to the previous experiments, with two key differences. First, we used the original trial timings (6 s, as in the original timing condition for Experiment 2). Second, all participants sat a single encoding and test phase, incorporating all 30 events. Missing responses were counted as incorrect ($M = 0.01$, s.d. $= 0.02$).

## 8.3. Analyses

Our primary aim for this experiment was to re-establish dependency in line with previous studies, in order to better understand how dependency is disrupted. Only a single one-sample $t$-test was required to test whether levels of dependency were significantly different from 0.

# 9. Experiment 3a results

## 9.1. Accuracy

Participants showed a mean proportion of 0.60 correct (s.d. $= 0.21$; figure 2$a$). This lower accuracy relative to Experiments 1 and 2 is in line with the increased memory demands of encoding all 30 events in a single experimental block.

## 9.2. Dependency

Dependency was still very low in this experiment ($M = 0.01$, s.d. $= 0.04$; figure 2$b$), and was not significantly different from zero ($t_{44} = 1.65$, $p = 0.106$, $d = 0.25$; $BF_{01} = 1.76$).

# 10. Experiment 3 interim summary

Despite the increased event numbers, Experiment 3a did not show evidence of dependency. While Experiments 2 and 3a have not shown strong evidence in favour of the null hypothesis, effect size estimates remain lower (range 0.21–0.28) than those in previous studies (range 0.5–1.26). In Experiment 3b, we turn to the hypothesis that stimulus format matters. In the majority of previous studies, stimuli have been presented as written words, asking participants to visualize the concepts interacting as vividly as possible. To remove reading ability as a source of variability for children in the original developmental study (Experiment 1), we simultaneously presented cartoon illustrations and spoken words. Doing so had not initially caused concern, given that Ngo *et al.* [14] had shown evidence of dependency with similar stimuli in a simultaneous encoding condition (replicated in Experiment 1). However, perhaps the format of the stimuli affects integration across encoding trials. In Experiment 3b, we tested whether retrieval dependency would re-emerge when presenting the same concepts as written words (https://osf.io/28v5u).

# 11. Experiment 3b methods

## 11.1. Participants

For the remaining experiments, we recruited 20 participants per condition. Experiment 1 had been powered to show evidence of dependency in the light of the increased variability and noise in developmental settings, and to enable robust tests of group differences. Given that the follow-up experiments have been much simpler in design, involving only a single condition, we re-powered these final experiments to preserve resources. We reviewed previous published experiments that tested for retrieval dependency immediately after encoding across separated pairs ($n = 10$ experiments), and computed an average effect size of $d = 0.86$ (weighted by sample size). We used the *pwr* package in R [32] to determine that a sample size of 20 would provide 95% power to detect a significant effect of this magnitude.

The 20 participants included in Experiment 3b included participants aged 19–34 years ($M = 28.39$ years). Twelve participants reported as female, seven male and one other. A further eight participants completed the study but were excluded from analyses on the basis of performance as in previous studies (seven floor, one ceiling), and seven participants were discontinued for failing the attention trials.

## 11.2. Design and procedure

The task was identical to Experiment 3a, with the exception that all stimuli were presented on screen as written words only (rather than pictures). That is, we used the same events constructed of animals, items and locations, but presented them in a different modality. Auditory presentation of the spoken words was also removed. As above, missing responses were counted as incorrect ($M = 0.04$, s.d. $= 0.05$).

# 12. Experiment 3b results

## 12.1. Accuracy

Participants showed a mean proportion of 0.63 correct (s.d. $= 0.18$), which was largely comparable to Experiment 3a (figure 2*a*).

## 12.2. Dependency

Levels of dependency were numerically higher ($M = 0.03$, s.d. $= 0.04$) relative to Experiment 3a ($M = 0.01$, s.d. $= 0.04$; figure 2*b*), and were significantly different from zero ($t_{19} = 3.28$, $p = 0.004$, $d = 0.73$; $BF_{01} = 0.09$).

## 12.3. Exploratory analyses

To test whether dependency was significantly higher when using written words compared with the picture and spoken word format, we carried out an additional exploratory *t*-test to compare dependency in Experiments 3a and 3b. There was no statistically significant difference between the two experiments ($t_{34.31} = 1.91$, $p = 0.064$, $d = 0.53$; $BF_{01} = 0.76$).

# 13. Experiment 3 discussion

In Experiment 3b, we found evidence of retrieval dependency when event elements were presented as written words (and not presented auditorily). Importantly, this provides key evidence that it is possible to measure retrieval dependency via an online platform, with an effect size comparable to those reported in previous laboratory-based experiments ($d = 0.73$; previous studies range from 0.5 to 1.26). While the difference in dependency between the two experiments with different stimulus formats was not significant, the pattern of results over the experiments presented implies that the inclusion of pictures and/or spoken words at least reduced dependency during retrieval. We, therefore, explored which of these elements was most problematic for participants binding across encoding trials.

# 14. Experiments 4a and 4b methods

In Experiment 4, we tested whether images or spoken words independently disrupt binding across separately encoded pairs. In the first instance, we predicted that the presentation of images would reduce dependency for the image condition relative to the spoken word condition. Experiment 4a (https://osf.io/zx47y) suffered an intermittent server issue on Gorilla that meant that—although we replaced participants who reported technical issues during the experiment ($n = 13$)—we could not be certain in the quality of the data. Experiment 4b (https://osf.io/ac9mb) thus sought to replicate the findings in an identical experiment.

## 14.1. Participants

### 14.1.1. Experiment 4a

We recruited 40 eligible participants, 20 in each of the stimulus format conditions (powered as Experiment 3b). A further 16 participants started the study but were discontinued for failing attention trials, and 14 completed all tasks but were excluded from analysis (six floor performance; seven ceiling performance and an additional participant who began the experiment after our pre-registered threshold was reached). The final sample had a mean age of 26.86 years (range 18–35 years). Twenty-six reported as female, 13 male and one other.

### 14.1.2. Experiment 4b

The final 40 participants for 4b had a mean age of 26.90 years (range 18–35 years), with 23 females and 17 males. Thirteen participants were discontinued for failing attention trials, and a further 13 were excluded from analyses on the basis of performance (seven floor, six ceiling).

## 14.2. Design and procedure

Experiments 4a and 4b used a between-subjects design to test for evidence of dependency in one of two stimulus format conditions: pictures versus words. The picture condition presented the cartoon images only, whereas the word condition presented both written and spoken words. Participants were randomly assigned to a condition at the start of the encoding phase until our target sample for one condition was met (with the final participants assigned to the remaining condition only). The trial timings and experimental set-up were otherwise identical to Experiments 3a and 3b. Missing responses were counted as incorrect (Experiment 4a: $M = 0.03$, s.d. $= 0.06$; Experiment 4b: $M = 0.04$, s.d. $= 0.07$).

# 15. Experiments 4a and 4b results

## 15.1. Experiment 4a results

### 15.1.1. Accuracy

Participants showed slightly higher memory performance for the picture condition ($M = 0.68$, s.d. $= 0.16$) compared with the word condition ($M = 0.60$, s.d. $= 0.22$; figure 2a). However, this difference was not statistically significant ($\beta = -0.34$, s.e. $= 0.33$; $Z = -1.03$; $p = 0.302$).

### 15.1.2. Dependency

We had initially predicted that pictures would be most disruptive to dependency, given the potential to interfere with visualization of the elements interacting. The level of dependency in this condition was relatively low ($M = 0.01$, s.d. $= 0.03$), with no statistical evidence that dependency in the data was significantly greater than 0 ($t_{19} = 2.00$, $p = 0.060$, $d = 0.45$; $BF_{01} = 0.83$). For the spoken word condition, there was very little evidence that dependency was greater than 0 ($M = 0.00$, s.d. $= 0.04$; $t_{19} = 0.30$, $p = 0.767$, $d = 0.07$; $BF_{01} = 4.13$), with the results favouring the null hypothesis. Finally, there was no significant difference in dependency between the two stimulus format conditions ($t_{36} = 1.00$, $p = 0.324$, $d = 0.32$; $BF_{01} = 2.18$; figure 2b).

## 15.2. Experiment 4b results

### 15.2.1. Accuracy

As in Experiment 4a, performance was slightly higher for the picture condition ($M = 0.66$, s.d. $= 0.17$) than the spoken word condition ($M = 0.62$, s.d. $= 0.18$; figure 2a), but this difference was not statistically significant ($\beta = -0.22$, s.e. $= 0.29$; $Z = -0.74$; $p = 0.458$).

### 15.2.2. Dependency

The pattern of results was very similar to Experiment 4a (figure 2b). The level of dependency in the picture condition was very low ($M = 0.02$, s.d. $= 0.04$) and not statistically different from 0 ($t_{19} = 1.73$, $p = 0.099$, $d = 0.39$; $BF_{01} = 0.82$). Again, there was no dependency in the spoken word condition ($M = 0.00$, s.d. $= 0.03$), with no significant difference from 0 ($t_{19} = 0.00$, $p = 0.998$, $d = 0.00$; $BF_{01} = 4.30$) and evidence in favour of the null hypothesis. The difference between the two conditions was not statistically significant ($t_{35.63} = 1.37$, $p = 0.178$, $d = 0.43$; $BF_{01} = 1.54$).

# 16. Exploratory analyses

## 16.1. Reduced dependency for picture presentation

In both Experiments 4a and 4b, it seemed clear that spoken words were disruptive to binding across encoding trials, with moderate evidence in favour of a null effect in each (i.e. no dependency). However, both showed 'marginal' evidence (though not statistically significant in either experiment) in favour of dependency in the picture presentation condition, suggesting that dependency may be present but with a lower effect size than in studies where written word presentation is used. As such, our smaller samples would have been under-powered to detect significant evidence of dependency. To inform future studies, we, therefore, carried out an additional exploratory analysis that combined both samples from Experiments 4a and 4b, providing better statistical power to test for dependency.

In the picture condition, there was evidence for dependency when the samples were combined ($t_{39} = 2.60$, $p = 0.013$, $d = 0.41$; $BF_{01} = 0.31$), albeit at a smaller effect size than reported in previous studies (range: 0.5–1.26). By contrast, dependency in the spoken word condition was not significant, with evidence in favour of the null hypothesis ($t_{39} = 0.23$, $p = 0.818$, $d = 0.04$; $BF_{01} = 5.72$). We entered the data into a 2 × 2 ANOVA that also incorporated experiment ID as an independent variable for reassurance that any residual technical problems in Experiment 4a were unlikely to have influenced the data. There was no difference in dependency between experiments ($p = 0.968$), and no interaction between experiment and stimulus format ($p = 0.725$). The difference in dependency between word and picture presentation was not significant, even with the greater statistical power of the combined samples ($F_{1,76} = 2.85$, $p = 0.095$).

## 16.2. Pictures versus written words

Given some evidence that dependency exists with picture presentation but potentially at a reduced level to previous experiments, we also tested whether there remained a significant difference in dependency between the picture condition (Experiments 4a, 4b) and written word experiment (Experiment 3b). Dependency was numerically higher in the written word condition ($M = 0.03$, s.d. $= 0.04$) than in the picture condition ($M = 0.01$, s.d. $= 0.04$), but this difference was not statistically significant ($t_{36.07} = 1.29$, $p = 0.204$, $d = 0.36$; $BF_{01} = 1.77$).

# 17. General discussion

We present the first experiments to document differences in dependency between simultaneously and separately encoded event elements, using a well-established paradigm. The evidence for retrieval dependency using the separated encoding paradigm in its typical format—with written word presentation—has been replicated many times across studies, and in Experiment 3b here. Thus, evidence for dependency in the separated encoding condition is highly replicable when stimuli are presented as written words at encoding and retrieval. However, we show that integrating across separately encoded event elements is affected by differences in stimulus format. In sum, we failed to find evidence of retrieval dependency across five experiments that used cartoon images and spoken words, rather than written words. In our final experiments (4a, 4b), we showed that the inclusion of spoken words was most problematic to binding across trials, but that cartoon images also produced smaller effects of dependency than have been documented in previous studies. That is, although previous studies have shown that participants can make inferences across overlapping trials in these different modalities (e.g. [33,34]), here we show that retrieval dependency, a more targeted measure of integration, is less likely to be seen. We discuss the conditions under which retrieval dependency is established or reduced, and speculate about the possible underlying mechanisms that underpin our ability to integrate overlapping information across separate encoding trials into a holistic memory representation.

Starting with conditions under which we observed clear evidence for retrieval dependency, this study contributes two key findings to the literature. First, Experiment 1 provided evidence that simultaneous encoding conditions are more robust to experimental changes than separated encoding conditions. That is, the changes we made in the current study relative to previous studies only disrupted dependency for separated encoding conditions. Our results replicate the findings of Ngo *et al.* [14] in showing that holistic retrieval of simultaneously encoded event elements can emerge when using cartoon-like stimuli. Importantly, we extended their findings to show clear evidence of dependency from simultaneously encoded event elements despite the inclusion of spoken words, reduced trial timings and fewer event numbers relative to previous experiments (e.g. [3]). Second, in Experiment 3b, we replicated earlier findings that holistic retrieval can emerge after separately encoding event elements when using written word stimulus presentation. Here, we showed that such effects can be successfully detected when collecting data *online*, and that this produced a similar effect size to previously published laboratory-based experiments ($d = 0.73$; laboratory range 0.5–1.26). Thus, dependency in separated encoding conditions appears highly replicable under specific experimental conditions (even when tested online), but less robust to changes in stimulus format.

Turning to conditions under which holistic retrieval does *not* emerge, we showed that participants were less likely to show retrieval dependency when required to bind the image and/or spoken word stimuli across trials (i.e. as opposed to written word stimuli). It is important to note that dependency is probably *reduced* rather than absent in these conditions: the difference in dependency between stimulus format conditions was not statistically significant in exploratory analyses (pictures versus written words; pictures versus words), and strong evidence in favour of the null hypothesis was rare across experiments. Further, collapsing across Experiments 4a and 4b revealed evidence for retrieval dependency when presenting visual images (in the absence of spoken words) at a smaller effect size to those documented in previous studies ($d = 0.41$). As such, while the present results may not be qualitatively different to previous research, we highlight that future studies using these alternative formats will probably require larger sample sizes to detect dependency within a condition or differences between conditions.

In Experiments 4a and 4b, we determined that the inclusion of spoken words (simultaneously presented with written words) appeared most detrimental to retrieval dependency, but that the use of images also weakened the effect relative to previous studies. We argue that this weakened dependency relates to integrative processes at encoding, given that an identical retrieval task was used to successfully capture retrieval dependency in the simultaneous condition of Experiment 1. At encoding, participants are required to imagine the two items interacting with each other, a process that is probably reliant on visual imagery. Previous evidence demonstrates that mental imagery as an encoding strategy, relative to rote repetition, increases overall memory performance [35], and this process might also be critical to the binding of information across separate encoding trials. We speculate on why the binding of information across trials might be affected by stimulus format, starting first with spoken words before considering the inclusion of pictures.

Though speculative, we propose three—not mutually exclusive—mechanisms by which the inclusion of spoken words might interfere with the binding of separated event elements. First, the inclusion of spoken words in these experiments was always alongside information in another modality (images, written words). The presentation of information across multiple modalities (visual and auditory) necessarily incorporates additional episodic information, which may better segregate the different encoding trials in memory and prevent their integration. By this account, spoken words alone may allow for greater integration than the simultaneous presentation of spoken and written words. Second, it may relate to the time participants are given to integrate across trials, whether that is achieved via visual imagery or verbal elaboration strategies. While Experiments 2–4b matched overall encoding trial time (6 s) to previous experiments [3,13,18,19], memory integration processes may have been disrupted during one-third of this time by the presentation of the spoken words. Thus, while our conditions matched *overall* trial times to previous studies, only 4 s remained for uninterrupted integration. Our manipulation of trial timings in Experiment 2 did not suggest timings to be important, but they may not have reached a critical threshold for participants to sufficiently imagine—and therefore integrate—across trials. This suggestion is supported by a study from Anderson & McCulloch [36], who found that increasing encoding time increased the likelihood that participants engaged in spontaneous integration when encoding category-related exemplars. This account would predict greater dependency under extended encoding conditions, reaching equivalent levels to previous studies when sufficient time is given. Third, the presentation of spoken words may have encouraged the use of more verbal strategies (e.g. repetition of the two words; creating a sentence) over and above more in-depth visualization strategies, which may be less effective at binding elements across separate encoding trials in episodic memory. This account would predict reduced dependency even in written word conditions under different encoding instructions, for example, mental imagery versus verbal rehearsal [35].

Although differences in dependency between picture and written word modalities were not statistically significant, it is still worth considering why the picture presentation conditions in Experiments 4a and 4b had substantially smaller effect sizes than in previously published research. Indeed, these differences may have important implications for future experimental design. In considering how pictures may disrupt the integration process, other studies have suggested that items presented as pictures remain more distinct in memory than those presented as words. For example, participants are less susceptible to falsely recognize lure items in the Deese–Roediger–McDermott paradigm following picture encoding of line drawings, compared with word presentation (e.g. [37]). The enhanced separation could be because pictures provide episodic details that prevent integration, or because the resulting visualizations are less vivid than those that have been internally generated and based on prior experience. Interestingly, reduced integration was not apparent in a study by Bisby *et al.* [13] that demonstrated holistic retrieval from separately encoded pairs of photographs (and at a comparable effect size to the rest of the literature). Perhaps then, photographs allow more realistic imagery of events akin with autobiographical experiences, whereas our cartoon-like images prevented such in-depth encoding. This explanation is entirely speculative at present, but would predict greater dependency if photographs were substituted into the present study.

We suggest that one way in which memory integration is disrupted in this paradigm is via interrupting mental imagery processes at encoding—in line with participants' instructions to visualize the elements interacting as vividly as possible. Why might mental imagery be critical to the binding of information across separate encoding trials? Episodic memory is typically defined in relation to an individual spatio-temporal event [1], yet integrating and generalizing across related events can help inform goal-directed behaviour [15,38]. Thus, in some situations, it is beneficial to integrate information across different spatio-temporal contexts. One possibility is that participants, on the third encoding trial, are explicitly bringing to mind the three related elements of an 'event' and creating a single mental image for all three elements—leading to behavioural dependency at retrieval. This would be consistent with the finding that the hippocampal BOLD response in the anterior hippocampus on the third encoding trial predicts memory performance for the pairwise associations learnt on the first and second encoding trial for that 'event' [18]. Though not mutually exclusive, a second possibility is that mental imagery provides a common 'context' representation across the three related encoding trials (akin to the context representation in the temporal context model; Howard & Kahana [39]), increasing the probability that retrieval of separately encoded information is related at retrieval. Though highly speculative, we believe that further research investigating the role of mental imagery in the encoding and integration of episodic information may help provide novel insight into these core mnemonic processes.

In conclusion, the present experiments show that additional processes at encoding are important for integrating separately—versus simultaneously—encoded event elements in episodic memory, and that these processes are affected by stimulus modality in experimental studies. Our ability to integrate separately encoded elements into a coherent event memory is optimal when cued by written words, and at least weakened—if not entirely disrupted—by alternative presentation formats. Direct investigations of the processes involved in memory integration (e.g. the role of visual imagery) are now needed to examine *how* they are affected by presentation modality. However, our findings highlight that caution is required when adapting the paradigm to address related questions of episodic memory, as these adaptations may require (at least) larger sample sizes to achieve the desired statistical power. From an applied perspective, the findings highlight that building memory structures from overlapping associations may have limited application in educational settings, and that robust memories of this nature are more reliably formed via simultaneous encoding of event elements within the same temporal context.

Ethics. All experiments were conducted in accordance with the British Psychological Society ethical guidelines, and were approved by the Psychology Research Ethics Committee at the University of York.

Data accessibility. Materials, data and analysis scripts can be accessed via the Open Science Framework page http://osf.io/cqm7v. The online experiments can be accessed via Gorilla Open Materials: https://gorilla.sc/openmaterials/59473.

Authors' contributions. E.J. contributed to the conception and design of each experiment, collected and analysed the data, and drafted the manuscript; G.O. contributed to data collection, and critically revised the manuscript; L.M.H. contributed to the conception and design of the study, and critically revised the manuscript; A.J.H. contributed to the conception and design of the study, the interpretation of the data and helped to draft the manuscript. All authors gave final approval for publication and agree to be held accountable for the work performed therein.

Competing interests. We declare we have no competing interests.

Funding. The research was funded by ESRC grant ES/R007454/1 awarded to A.J.H. and L.M.H. E.J. was additionally supported by ESRC fellowship ES/T007524/1, and L.M.H. by ESRC grant no. ES/N009924/1.

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
