## [Reviewer comments · Royal Society Open Science]

Review History

RSOS-200431.R0 (Original submission)

Review form: Reviewer 1

Is the manuscript scientifically sound in its present form?

Yes

Are the interpretations and conclusions justified by the results?

No

Is the language acceptable?

Yes

Do you have any ethical concerns with this paper?

No

Have you any concerns about statistical analyses in this paper?

No

Recommendation?

Reject

Comments to the Author(s)

James and colleagues report a series of experiments investigating the boundary conditions of memory dependency when related pairs are encoded across separate experiences. The main finding is that while there is robust memory dependency when three stimuli are presented simultaneously, while retrieval dependency of separately encoded events is less reliable and largely depends on stimulus presentation format. The authors draw the conclusion that under separate encoding conditions, holistic retrieval was only found for written words, whereas cartoon pictures and spoken words both weakened the integration across events. (They also found that contrary to their predictions, extend encoding time did not increase dependency.)

This work extends upon the authors' previous findings of retrieval dependency for separately encoded associations to provide some characterization of the encoding factors that dictate (or enhance/inhibit) integration. It is an interesting topic, and I appreciate the attempts to systematically isolate those encoding factors that might matter with empirical data. However, I do have concerns about the paper and the authors' interpretation of the data, which I detail below.

Major

1) Throughout the paper, I felt the authors were at times speaking well beyond their data. For example, the main interpretation of the results is with regards to the spoken words disrupting mental imagery; however, they have no data to support this (mental imagery disruption) as a mechanism. It seems to me a big leap to go from the main finding that spoken words interfere with dependency in the separate encoding situation to visual imagery is important for integration and its disruption is the problem. Mental imagery measures were not included in any of the experiments, so it is impossible to know whether different levels of retrieval dependency resulted from mental imagery differences during encoding. There is simply no way of knowing whether the auditory stimuli (or some other condition difference) interfered with visual imagery or something else. Unless the authors have data to speak to this (e.g., if additional analyses can be performed to bolster this claim), I would recommend they remove the interpretation from the paper.

2) Given the data, I felt there was far too much focus on the disruption of integration caused by pictures (vs. written words). I understand that the effects are numerically reduced and are showing only trend-level dependency rather than significant; however, the pictures vs. words comparison is not significant (p. 24). Thus, any conclusions that rest upon a difference between these conditions are unfounded. For example, in the discussion, the authors write that "but that cartoon images also reduced retrieval dependency relative to previous studies" (p. 25) but I am not sure to what statistical test this is referring? It says there was no difference in dependency between picture only (4a/b) and written only (3b) experiments (p. 24). There is also text alluding to picture-related decreases in the abstract.

3) Many references to previous related work showing integration across experiences at behavioral and/or neural levels were noticeably missing from the paper. One issue in particular that came to mind was the disconnect with prior work in terms of stimulus materials; many of this previously published work uses pictures and yet shows evidence for integration (e.g., Richter et al. (2016); Tompary & Davachi (2017); Schlichting et al. 2015; Zeithamova et al. (2012); Gershman et al. (2013); Milivojevic et al. (2015); Shohamy & Wagner (2008)). Some discussion reconciling the present conclusions with this large body of prior work is needed. In addition, there is a large body of literature about task differences and/or individual differences that seem to impact the degree of integration (some examples: Horton & Kjeldergaard (1961); Postman (1962); McCloskey & Bigler (1980); Moeser (1977); Radvansky & Zacks (1991); Anderson & McCulloch (1999); Hupbach et al. (2007); Gershman et al. (2013); Ellenbogen et al. (2013); Schlichting et al. (2015); Zeithamova & Preston (2017); Tompary & Davachi (2017); Cai et al. (2016); Robin & Olsen (2019)).

When discussing the possibilities for boundary conditions, it would be important to more thoroughly characterize the existing literature on the topic.

4) A general weakness of the paper is that because not all combinations of the stimulus types were tested, it is a challenge to know specifically the cause of the drop in dependency. For example, the authors did not assess dependency for stimuli that were spoken words on their own (without written words or pictures). We also do not know how the written word + pictures together would fare. Dependency is lower in the cases when there are two stimuli (written words/pictures along with spoken words) vs. just one (pictures or words only), which the authors attribute to the spoken words; however, could it be about two vs. one stimulus per associate? This seems equally likely to the interpretation offered by the authors about spoken words being the culprit, since we do not know how dependency looks for written words + pictures or spoken words on their own.

5) For experiment 2, the authors seem to be drawing a relatively strong conclusion of no dependency in the 3s condition when the p value here is 0.066. It may very well be that (as the authors say) dependency does not depend on timing, but as a secondary point here it is notable that the 3s condition shows a trend for dependency which feels like very different from the conclusions of experiment 1. More text on why the two experiments might not be producing the same effects (e.g., in the experiment 2 discussion) is warranted.

6) It is stated in the general discussion that the authors "replicated" findings from Ngo et al. (2019) in experiment 1. However, aren't these the same data (e.g., would dependency in adults be in the present study's experiment 1, simultaneous condition be the same as the dark blue box plot in Figure 5B of Ngo et al. 2019)? Perhaps I am missing something because the numbers don't seem to match up exactly. The authors should clarify this and if it is a re-analysis of existing data that has been published, refrain from using the term "replication."

7) Implicit in much of the discussion text is a comparison between the simultaneous and separate encoding conditions (in terms of e.g., robustness to changes in stimulus format; p. 25). However, I believe the authors only tested the simultaneous condition with pictures in the present study; please clarify upon which result this claim is based.

8) One explanation provided for why spoken words might disrupt integration is that the words take some time to happen, thus reducing encoding time. The logic put forth is that when spoken words are present, greater dependency should be found when extending encoding time (p. 27); however, this seems to contradict what was found in experiment 2. Please clarify.

9) More details about the specific nature of the stimuli would be helpful, and whether they were matched between the word and picture versions of the experiment. For example, within (3a vs. 3b) or between experiments (3 vs. 4), were the written words the same as the pictures (across participants)? If they were different, how can we conclude the differences observed are due to format rather than content?

10) The authors might consider whether a statistical test that asks whether the presence of written words, spoken words, and pictures as three separate factors is significantly contributing to differences in dependency observed across experiments. In other words, rather than code by experiment, why not ask whether the presence of each of these three stimulus features is important?

11) I am not sure of the relevance of all the discussion about pictures (vs. cartoons vs. words) given none of the differences related to cartoons vs. words in the present study were reliable, and the idea about cartoons vs. photographs was not tested. The authors' ideas about the relationship between vividness/detail and integration was also not clear and feels beyond the scope of the data. Why might written (but not spoken) words be more likely to elicit visualization that shares

common features with real life experiences, and why would cartoon pictures have less access to this representation? Please clarify or remove this if it is beyond the scope of this study.

Minor

12) It was a bit difficult to keep track of what varied across all experiments. It might be helpful for readers to underscore these differences either on the graphs or in a table/figure.

13) Were there order effects for experiment 1? Participants knowledge of the task structure (expecting groups of three items) might influence the degree to which dependency is observed.

14) Was there a difference in accuracy between the separated condition in experiment 1 vs. 2? One possibility is that this apparently lower performance has to do with the online nature of the task or associated sample differences; alternatively, it might be that there are performance differences because these participants were never exposure to a "simultaneous" condition that highlights the groups of three in the task. Within experiment 2 I see that 6s>3s accuracy, but this seems to be driven mostly by a drop in the 3s condition compared with the analogous condition in experiment 1, and some speculation on why this might be would be useful.

15) Just a note that an additional explanation not mentioned by the authors in the discussion is that the verbal stimuli might interfere with participants' ability to imagine the objects interacting (especially for those participants who might naturally take more verbal strategies in imagining such a scenario). In other words they are already processing verbal information and cannot process more for the imagery encoding task. I am not sure if this is quite the same as the encoding time hypothesis, but I found it strange that this possibility was not mentioned since it seems plausible given working memory theories and literature.

Review form: Reviewer 2 (Jeffrey Starns)

Is the manuscript scientifically sound in its present form?

Yes

Are the interpretations and conclusions justified by the results?

Yes

Is the language acceptable?

Yes

Do you have any ethical concerns with this paper?

No

Have you any concerns about statistical analyses in this paper?

Yes

Recommendation?

Accept with minor revision (please list in comments)

Comments to the Author(s)

The authors test different variations on a paradigm for exploring binding of multidimensional event representations, realized here as events that combine an animal, location, and object. A critical manipulation in past studies was simultaneous versus separate encoding (i.e., study all three elements together in one episode or study pairs of elements in different episodes), and these studies show dependencies in retrieval of the elements in both cases (e.g., if a participant can

remember the location given the animal, then it is more likely that they can remember the object given the location). These studies focus mostly on the separate paradigm and show that the retrieval dependency breaks down in some conditions, like when the study items are both seen as a picture and heard as a word.

The experiments seem to be carefully designed and well powered, and I certainly appreciate the use a pre-registration. (A little more detail on the pre-registration could be useful, like whether predictions were part of the pre-registration and how they were expressed.) I didn't see a compelling theoretical motivation, but I'd be willing to be talked out of that impression by a revision. A deeper concern is that the paper attempts to distinguish procedures in terms of whether the produce significant or nonsignificant dependency, but even the "significant" ones seem like they show a very small effect. Put this together with the arbitrary definition of "significant" and the difficulty establishing that there is "no effect" in certain circumstances (i.e., the dependency is literally zero at the population level), and the implications get murky. I commend the inclusion of Bayesian statistics, but I think Bayesian credible intervals might be more instructive than Bayes Factors in the current context. A revision could better explain what is meaningful about the difference between a very small effect and "no" effect in this context (or show that I have misinterpreted the results by deeming them very small effects), so this could come down to an issue of writing. I also need some convincing that the dependency measure is a good one. So in summary, I think the authors should be given a chance to revise, and I would like to see the revision clarify some issues and maybe take a different analysis approach. I will explain my recommendations below.

Let me first say that it was difficult for me to track all of the threads in the different procedures and results. This is mostly because the authors have very thoroughly explored the possibilities with an impressive array of studies, so that is a positive, of course. However, I would love to see a revision with one big figure or table summarizing all the results together, with a clear demarcation for which experiments fall on either side of the distinction that the authors are claiming is the key to getting or not getting dependency (color coding or something). I would like to see confidence intervals in this big figure, as in the figures already provided, but it might get too squished to try to put in all the data points.

One of the biggest issues to address is clarifying the theoretical contribution. There is some theoretical speculation in the General Discussion, but I missed it earlier in the paper. Most critically, I was often wondering why people cared about the simultaneous/separate distinction and whether the current results have any new implications for those issues. Has the fact that you see dependency in separate encoding conditions been used to support any specific theoretical conclusions, and would these conclusions change based on observing that the dependency goes away with auditory presentation at encoding? There are some hints at these issues, but I think the picture could be filled in a lot more.

I thought the paper could have said more about the rationale for the specific encoding procedures. Specifically, I'm sure there were efforts to try to ensure that each individual item had the same learning opportunity in the simultaneous and separate conditions, but it didn't seem like there was a clear statement of the strategy for achieving this.

The dependency measure did not seem to be strongly theoretically motivated. Are there any demonstrations that it has good measurement properties; i.e., that it consistently tracks the dependency of memory representations without being influenced by other factors? Do we know the sampling distribution for this measure?

"For the spoken word condition, there was very little evidence that dependency was greater than 0 ($M = .00$, $SD = .04$; $t(19) = 0.30$, $p = .767$, $d = 0.07$; $BF_{01} = 4.13$)." The Bayes Factor indicates that there was no evidence at all for dependency; indeed, there was moderate evidence *against* dependency (i.e., evidence for the null).

Were participants with performance near ceiling excluded because it is difficult to tell whether or not they show retrieval dependency? Were the criteria for excluding these participants described in the pre-registration for the pre-registered studies? If not, it is a good idea to note whether any conclusions change when they are included.

Jeff Starns

Decision letter (RSOS-200431.R0)

Dear Dr James,

The editors assigned to your paper ("Make or break it: Boundary conditions for integrating multiple elements in episodic memory") have now received comments from reviewers. We would like you to revise your paper in accordance with the referee and Associate Editor suggestions which can be found below (not including confidential reports to the Editor). Please note this decision does not guarantee eventual acceptance.

Please submit a copy of your revised paper before 04-Jun-2020. Please note that the revision deadline will expire at 00.00am on this date. If we do not hear from you within this time then it will be assumed that the paper has been withdrawn. In exceptional circumstances, extensions may be possible if agreed with the Editorial Office in advance. We do not allow multiple rounds of revision so we urge you to make every effort to fully address all of the comments at this stage. If deemed necessary by the Editors, your manuscript will be sent back to one or more of the original reviewers for assessment. If the original reviewers are not available, we may invite new reviewers.

- Data accessibility

<http://datadryad.org/submit?journalID=RSOS&manu=RSOS-200431>

- Competing interests

- Authors' contributions

- Acknowledgements

- Funding statement

Kind regards,

Andrew Dunn

on behalf of Dr Alexa Morcom (Associate Editor)

We have received two reviews of your paper, "Make or break it: Boundary conditions for integrating multiple elements in episodic memory" (RSOS-200431). Although the reviewers feel that these results are of potential interest, they have raised a number of substantive concerns, which preclude publication of the paper in Royal Society Open Science, at least in its present form.

Both reviewers question the interpretation that varying levels of apparent dependency reflect the importance of visual imagery in printed word more than in picture or spoken words conditions. The discussion and conclusions should be better balanced in this regard. The paper would also be strengthened if further, exploratory analyses could more directly support this story.

There are also concerns about the statistical inference. Reviewer 2 believes that the results do not always support the inferences drawn in terms of which conditions differ, which is a serious concern. Reviewer 1 queries interpretation of, and evidence for, small effect sizes as opposed to 'no effect'. It may be that this can be remedied by additional analyses directly supporting the inferences (Reviewer 2) or by stronger Bayesian inference, by use of credible intervals (Reviewer 1) or at least some Bayes Factor robustness checks.

The reviews are appended to this message. We hope that you will be able to address the reviewers' concerns in full and resubmit the manuscript, along with a point-by-point reply to the reviews that indicates your response to each concern. Before we make a decision about publication, we will have your revision re-reviewed by the reviewers.

Sincerely,

Alexa Morcom (Action Editor)

Comments to Author:

Reviewers' Comments to Author:

Reviewer: 1

Comments to the Author(s)

James and colleagues report a series of experiments investigating the boundary conditions of memory dependency when related pairs are encoded across separate experiences. The main finding is that while there is robust memory dependency when three stimuli are presented simultaneously, while retrieval dependency of separately encoded events is less reliable and largely depends on stimulus presentation format. The authors draw the conclusion that under separate encoding conditions, holistic retrieval was only found for written words, whereas cartoon pictures and spoken words both weakened the integration across events. (They also found that contrary to their predictions, extend encoding time did not increase dependency.)

This work extends upon the authors' previous findings of retrieval dependency for separately encoded associations to provide some characterization of the encoding factors that dictate (or enhance/inhibit) integration. It is an interesting topic, and I appreciate the attempts to systematically isolate those encoding factors that might matter with empirical data. However, I do have concerns about the paper and the authors' interpretation of the data, which I detail below.

Major

1) Throughout the paper, I felt the authors were at times speaking well beyond their data. For example, the main interpretation of the results is with regards to the spoken words disrupting mental imagery; however, they have no data to support this (mental imagery disruption) as a mechanism. It seems to me a big leap to go from the main finding that spoken words interfere with dependency in the separate encoding situation to visual imagery is important for integration and its disruption is the problem. Mental imagery measures were not included in any of the experiments, so it is impossible to know whether different levels of retrieval dependency resulted from mental imagery differences during encoding. There is simply no way of knowing whether the auditory stimuli (or some other condition difference) interfered with visual imagery or

something else. Unless the authors have data to speak to this (e.g., if additional analyses can be performed to bolster this claim), I would recommend they remove the interpretation from the paper.

2) Given the data, I felt there was far too much focus on the disruption of integration caused by pictures (vs. written words). I understand that the effects are numerically reduced and are showing only trend-level dependency rather than significant; however, the pictures vs. words comparison is not significant (p. 24). Thus, any conclusions that rest upon a difference between these conditions are unfounded. For example, in the discussion, the authors write that "but that cartoon images also reduced retrieval dependency relative to previous studies" (p. 25) but I am not sure to what statistical test this is referring? It says there was no difference in dependency between picture only (4a/b) and written only (3b) experiments (p. 24). There is also text alluding to picture-related decreases in the abstract.

3) Many references to previous related work showing integration across experiences at behavioral and/or neural levels were noticeably missing from the paper. One issue in particular that came to mind was the disconnect with prior work in terms of stimulus materials; many of this previously published work uses pictures and yet shows evidence for integration (e.g., Richter et al. (2016); Tomparý & Davachi (2017); Schlichting et al. 2015; Zeithamova et al. (2012); Gershman et al. (2013); Milivojevic et al. (2015); Shohamy & Wagner (2008)). Some discussion reconciling the present conclusions with this large body of prior work is needed. In addition, there is a large body of literature about task differences and/or individual differences that seem to impact the degree of integration (some examples: Horton & Kjeldergaard (1961); Postman (1962); McCloskey & Bigler (1980); Moeser (1977); Radvansky & Zacks (1991); Anderson & McCulloch (1999); Hupbach et al. (2007); Gershman et al. (2013); Ellenbogen et al. (2013); Schlichting et al. (2015); Zeithamova & Preston (2017); Tomparý & Davachi (2017); Cai et al. (2016); Robin & Olsen (2019)). When discussing the possibilities for boundary conditions, it would be important to more thoroughly characterize the existing literature on the topic.

4) A general weakness of the paper is that because not all combinations of the stimulus types were tested, it is a challenge to know specifically the cause of the drop in dependency. For example, the authors did not assess dependency for stimuli that were spoken words on their own (without written words or pictures). We also do not know how the written word + pictures together would fare. Dependency is lower in the cases when there are two stimuli (written words/pictures along with spoken words) vs. just one (pictures or words only), which the authors attribute to the spoken words; however, could it be about two vs. one stimulus per associate? This seems equally likely to the interpretation offered by the authors about spoken words being the culprit, since we do not know how dependency looks for written words + pictures or spoken words on their own.

5) For experiment 2, the authors seem to be drawing a relatively strong conclusion of no dependency in the 3s condition when the p value here is 0.066. It may very well be that (as the authors say) dependency does not depend on timing, but as a secondary point here it is notable that the 3s condition shows a trend for dependency which feels like very different from the conclusions of experiment 1. More text on why the two experiments might not be producing the same effects (e.g., in the experiment 2 discussion) is warranted.

6) It is stated in the general discussion that the authors "replicated" findings from Ngo et al. (2019) in experiment 1. However, aren't these the same data (e.g., would dependency in adults be in the present study's experiment 1, simultaneous condition be the same as the dark blue box plot in Figure 5B of Ngo et al. 2019)? Perhaps I am missing something because the numbers don't seem to match up exactly. The authors should clarify this and if it is a re-analysis of existing data that has been published, refrain from using the term "replication."

7) Implicit in much of the discussion text is a comparison between the simultaneous and separate encoding conditions (in terms of e.g., robustness to changes in stimulus format; p. 25). However, I

believe the authors only tested the simultaneous condition with pictures in the present study; please clarify upon which result this claim is based.

8) One explanation provided for why spoken words might disrupt integration is that the words take some time to happen, thus reducing encoding time. The logic put forth is that when spoken words are present, greater dependency should be found when extending encoding time (p. 27); however, this seems to contradict what was found in experiment 2. Please clarify.

9) More details about the specific nature of the stimuli would be helpful, and whether they were matched between the word and picture versions of the experiment. For example, within (3a vs. 3b) or between experiments (3 vs. 4), were the written words the same as the pictures (across participants)? If they were different, how can we conclude the differences observed are due to format rather than content?

10) The authors might consider whether a statistical test that asks whether the presence of written words, spoken words, and pictures as three separate factors is significantly contributing to differences in dependency observed across experiments. In other words, rather than code by experiment, why not ask whether the presence of each of these three stimulus features is important?

11) I am not sure the relevance of all the discussion about pictures (vs. cartoons vs. words) given none of the differences related to cartoons vs. words in the present study were reliable, and the idea about cartoons vs. photographs was not tested. The authors' ideas about the relationship between vividness/detail and integration was also not clear and feels beyond the scope of the data. Why might written (but not spoken) words be more likely to elicit visualization that shares common features with real life experiences, and why would cartoon pictures have less access to this representation? Please clarify or remove this if it is beyond the scope of this study.

Minor

12) It was a bit difficult to keep track of what varied across all experiments. It might be helpful for readers to underscore these differences either on the graphs or in a table/figure.

13) Were there order effects for experiment 1? Participants knowledge of the task structure (expecting groups of three items) might influence the degree to which dependency is observed.

14) Was there a difference in accuracy between the separated condition in experiment 1 vs. 2? One possibility is that this apparently lower performance has to do with the online nature of the task or associated sample differences; alternatively, it might be that there are performance differences because these participants were never exposure to a "simultaneous" condition that highlights the groups of three in the task. Within experiment 2 I see that 6s>3s accuracy, but this seems to be driven mostly by a drop in the 3s condition compared with the analogous condition in experiment 1, and some speculation on why this might be would be useful.

15) Just a note that an additional explanation not mentioned by the authors in the discussion is that the verbal stimuli might interfere with participants' ability to imagine the objects interacting (especially for those participants who might naturally take more verbal strategies in imagining such a scenario). In other words they are already processing verbal information and cannot process more for the imagery encoding task. I am not sure if this is quite the same as the encoding time hypothesis, but I found it strange that this possibility was not mentioned since it seems plausible given working memory theories and literature.

Reviewer: 2

Comments to the Author(s)

The authors test different variations on a paradigm for exploring binding of multidimensional event representations, realized here as events that combine an animal, location, and object. A critical manipulation in past studies was simultaneous versus separate encoding (i.e., study all three elements together in one episode or study pairs of elements in different episodes), and these studies show dependencies in retrieval of the elements in both cases (e.g., if a participant can remember the location given the animal, then it is more likely that they can remember the object given the location). These studies focus mostly on the separate paradigm and show that the retrieval dependency breaks down in some conditions, like when the study items are both seen as a picture and heard as a word.

The experiments seem to be carefully designed and well powered, and I certainly appreciate the use a pre-registration. (A little more detail on the pre-registration could be useful, like whether predictions were part of the pre-registration and how they were expressed.) I didn't see a compelling theoretical motivation, but I'd be willing to be talked out of that impression by a revision. A deeper concern is that the paper attempts to distinguish procedures in terms of whether the produce significant or nonsignificant dependency, but even the "significant" ones seem like they show a very small effect. Put this together with the arbitrary definition of "significant" and the difficulty establishing that there is "no effect" in certain circumstances (i.e., the dependency is literally zero at the population level), and the implications get murky. I commend the inclusion of Bayesian statistics, but I think Bayesian credible intervals might be more instructive than Bayes Factors in the current context. A revision could better explain what is meaningful about the difference between a very small effect and "no" effect in this context (or show that I have misinterpreted the results by deeming them very small effects), so this could come down to an issue of writing. I also need some convincing that the dependency measure is a good one. So in summary, I think the authors should be given a chance to revise, and I would like to see the revision clarify some issues and maybe take a different analysis approach. I will explain my recommendations below.

Let me first say that it was difficult for me to track all of the threads in the different procedures and results. This is mostly because the authors have very thoroughly explored the possibilities with an impressive array of studies, so that is a positive, of course. However, I would love to see a revision with one big figure or table summarizing all the results together, with a clear demarcation for which experiments fall on either side of the distinction that the authors are claiming is the key to getting or not getting dependency (color coding or something). I would like to see confidence intervals in this big figure, as in the figures already provided, but it might get too squished to try to put in all the data points.

One of the biggest issues to address is clarifying the theoretical contribution. There is some theoretical speculation in the General Discussion, but I missed it earlier in the paper. Most critically, I was often wondering why people cared about the simultaneous/separate distinction and whether the current results have any new implications for those issues. Has the fact that you see dependency in separate encoding conditions been used to support any specific theoretical conclusions, and would these conclusions change based on observing that the dependency goes away with auditory presentation at encoding? There are some hints at these issues, but I think the picture could be filled in a lot more.

I thought the paper could have said more about the rationale for the specific encoding procedures. Specifically, I'm sure there were efforts to try to ensure that each individual item had the same learning opportunity in the simultaneous and separate conditions, but it didn't seem like there was a clear statement of the strategy for achieving this.

The dependency measure did not seem to be strongly theoretically motivated. Are there any demonstrations that it has good measurement properties; i.e., that it consistently tracks the

dependency of memory representations without being influenced by other factors? Do we know the sampling distribution for this measure?

“For the spoken word condition, there was very little evidence that dependency was greater than 0 ($M = .00$, $SD = .04$; $t(19) = 0.30$, $p = .767$, $d = 0.07$; $BF_{01} = 4.13$).” The Bayes Factor indicates that there was no evidence at all for dependency; indeed, there was moderate evidence *against* dependency (i.e., evidence for the null).

Were participants with performance near ceiling excluded because it is difficult to tell whether or not they show retrieval dependency? Were the criteria for excluding these participants described in the pre-registration for the pre-registered studies? If not, it is a good idea to note whether any conclusions change when they are included.

Jeff Starns

Author's Response to Decision Letter for (RSOS-200431.R0)

See Appendix A.

RSOS-200431.R1 (Revision)

Review form: Reviewer 1

Is the manuscript scientifically sound in its present form?

Yes

Are the interpretations and conclusions justified by the results?

No

Is the language acceptable?

Yes

Do you have any ethical concerns with this paper?

No

Have you any concerns about statistical analyses in this paper?

Yes

Recommendation?

Reject

Comments to the Author(s)

The authors have generally been responsive to my previous comments in this revision. In particular, I appreciate that they removed interpretations regarding visual imagery and the like that were well beyond their data. The paper is now far less speculative. It also really made a difference for me that now when their claims are based on direct comparisons to published work (e.g., effect sizes), that is explicitly stated. My remaining comments are as follows.

1) Unfortunately, my main concern is one that I am not sure can be adequately addressed at this stage. As noted by the authors in their response, the paper is not really meant to make a theoretical contribution (which is fine). However, it's also not systematic enough to be methodologically very informative, either. Right now each experiment feels like a variation on the last to see what happens to the effect when you turn a knob; however, what would have been more useful would be a series of studies in which the authors systematically manipulate factors that might matter in, for example, a fully crossed fashion (say, duration, simultaneous/separate, load [number of events], written/spoken/pictures/cartoons, 1 vs. 2 modalities). Then, I think, you could draw conclusions about which factors matter and which do not. After reading this paper, I am still uncertain of the authors' conclusions because there seem to be many possible explanations for why effects may have changed across their studies (within this paper, as well as from previous work) because there are multiple differences.

2) With respect to my earlier suggestion to add more references to the previous literature, the authors have added some text but only discussing a very small number of these papers. To be clear, I am not wanting the authors add every one of my paper suggestions to their list. However, I still do think the treatment of prior work is a bit incomplete in their paper and I ask the authors to reconsider this issue. The authors seem to have rejected considering some of these papers because they did not investigate retrieval dependency per se or that there were (in my opinion) minor differences in the paradigm, even though many of them did investigate evidence of memory-to-memory connections (sure, sometimes without training all possible pairings) at the psychological and/or neural representational level. (To be clear, I am not necessarily referring to the inference test, which I agree is a bit different.) Surely despite these surface differences in task the mechanisms are the same or similar as what the authors investigate here. So, when the authors discuss how stimulus type/materials (pictures, cartoons, written words, spoken words), and spend paragraphs speculating about this in the discussion, it feels odd to ignore all of the papers that have demonstrated evidence for integration and discuss the nature of the stimuli in those studies. If the authors feel that for some reason what is going on in their task is fundamentally different from all of the published work using varied stimulus types (or that for the purposes of their paper they are defining "integration" in the narrow sense as being about retrieval dependency specifically), they need to address this directly in their paper.

3) Many of the conclusions are based on effect size comparisons versus prior work. Some of these conclusions seem fairly central to the current paper. The authors should be cautious and transparent in how they are making these comparisons. For example, it seems that comparing published effect sizes to those from the cartoon version of experiment 4b with published work could be problematic, given there are potentially myriad differences across the experiments. In other words, how can we know this is attributable to the cartoon images versus any of the other changes (including at least, I assume, they were collected at different times)? To compare the effect sizes is one thing, but to attribute the drop in effect size to a particular cause seems something else.

4) Some of the phrasing in the discussion is still speaking beyond the data, for example:
 - "In conclusion, the present experiments show that additional processes at encoding are important for integrating separately – versus simultaneously – encoded event elements in episodic memory, and that these processes are affected by stimulus modality in experimental studies." This is very speculative and does not necessarily follow from their data. I would agree that special mechanisms at encoding are important for integrating separately presented events, but I think we know that from other studies, not the present data. Do the authors think that separate but not simultaneous encoding is impacted by stimulus modality? Did they demonstrate this? I do not believe this was shown in the current study, so this may again be based on comparing effect sizes (and lack of dependency in some cases) to previous work (see comment 3).
 - "From an applied perspective, the findings highlight that building memory structures from overlapping associations may have limited application in educational settings, and that robust memories of this nature are more reliably formed via simultaneous encoding of event elements within the same temporal context." The meaning here is a bit unclear to me but is in my read

quite a stretch, even for a concluding sentence. Why is this the case (particularly in educational settings)? Do the authors mean to suggest that people cannot learn cross-event relationships in the classroom (or is this meant to be a comment about written vs. pictorial material)?

5) Please clarify which manipulations are within subject versus between subject in the table and/or graphs. It is unclear for example that in Exp 1 the simultaneous and separated encoding condition was a within subject manipulation, so I believe the $n=45$ is really just one group of participants for both conditions (and thus in a way, should not really be listed twice in the table).

6) In the version of the table in which changes are tracked in red underlined text (p. 48), there is an error such that all conditions have asterisks indicating significance. I believe the correct version of the table is the one without changes tracked (p. 8).

Review form: Reviewer 2 (Jeffrey Starns)

Is the manuscript scientifically sound in its present form?

Yes

Are the interpretations and conclusions justified by the results?

Yes

Is the language acceptable?

Yes

Do you have any ethical concerns with this paper?

No

Have you any concerns about statistical analyses in this paper?

Yes

Recommendation?

Accept as is

Comments to the Author(s)

The authors were considerate in responding to my suggestions for revision. I appreciate the added table with results across all experiments. I also appreciate the few changes to statistical practices, although I am disappointed that the authors did not choose to adopt a Bayesian parameter estimation approach. My concerns about the measurement properties of the dependency measure were not addressed, but the authors now emphasize that other people have used the same measure. I wonder if any of those people have explored the statistical properties of this measure (sampling distributions, recovery simulations, etc.; note that the distributions reported in the response to reviews are not sampling distributions). If so, that would be good to cite. If not, then just noting that other people used the measure before is not that comforting, but it is enough to meet typical publication standards. I also noted that the theoretical contribution was unclear, and the authors responded by clarifying that the studies are presented as a purely empirical contribution. This is fine as long as that is appropriate for the journal.

Jeff Starns

Review form: Reviewer 3 (Rose Cooper)

Is the manuscript scientifically sound in its present form?

Yes

Are the interpretations and conclusions justified by the results?

Yes

Is the language acceptable?

Yes

Do you have any ethical concerns with this paper?

No

Have you any concerns about statistical analyses in this paper?

No

Recommendation?

Accept with minor revision (please list in comments)

Comments to the Author(s)

In this paper, the authors present six experiments exploring the boundary conditions of holistic episodic memory retrieval after encoding separate, overlapping associations. This collection of experiments represents a thorough test of task factors that influence memory dependency in this condition, showing that presenting items as written words produces dependent memories, with associations presented as cartoons or spoken words apparently reducing dependency.

Overall, I found the methods and analyses to be robust and presented clearly. While I agree with previous reviewers that some discussion of task differences is complicated by non-significant statistical comparisons, the authors appear to have sufficiently addressed those concerns. The only additional comment I have concerns the stimulus modality vs. multimodal explanation of task differences – is there a reason that the authors chose not to systematically explore this question in their experiments? It seems that it would be just as important for future researchers to know whether to present stimuli in one (e.g. spoken only) vs. two modalities (e.g. written and spoken only) as it is to know which kind of stimulus modality to use.

Aside from the specific experimental manipulations, my main comment is about the theoretical importance of the central task. I appreciate that this is a methodological paper (one which will be informative for researchers implementing similar task designs), and the paper need not have theoretical importance in a direct sense. However, I would like the Introduction, and perhaps Discussion, to clarify why it is important to study the integration of overlapping associations from separate encoding trials, specifically. The authors note in the Introduction (page 4, lines 24-30) that the 'separate' condition was originally developed to avoid possible attention confounds with simultaneous presentation. Is this the only/primary benefit of a 'separate' task design, or does it have its own theoretical basis? In other words, simultaneous displays may represent single-shot episodic memories, but what do the separately-encoded associations uniquely capture? Does encoding associations on separate trials reflect the formation of a single episodic memory? Theoretical relevance is briefly alluded to on page 5 (lines 54-59), but further elaboration would be helpful. I think it is important for readers to come away from this paper knowing both how they should implement this task, and also why they should use it.

Decision letter (RSOS-200431.R1)

Dear Dr James

On behalf of the Editors, we are pleased to inform you that your Manuscript RSOS-200431.R1 "Make or break it: Boundary conditions for integrating multiple elements in episodic memory" has been accepted for publication in Royal Society Open Science subject to minor revision in accordance with the referees' reports. Please find the referees' comments along with any feedback from the Editors below my signature.

Please submit your revised manuscript and required files (see below) no later than 7 days from today's (ie 11-Aug-2020) date. Note: the ScholarOne system will 'lock' if submission of the revision is attempted 7 or more days after the deadline. If you do not think you will be able to meet this deadline please contact the editorial office immediately.

on behalf of Dr Alexa Morcom (Associate Editor)
openscience@royalsociety.org

Associate Editor Comments to Author (Dr Alexa Morcom):

Dear Dr James

Thank you for submitting the revised version of your paper, "Make or break it: Boundary conditions for integrating multiple elements in episodic memory" (RSOS-200431). Following a further round of review, I am pleased to inform you that we now can accept the paper subject to only minor revisions. The reviewers and I appreciate your responsiveness to their earlier suggestions.

The initial two reviewers both looked at the paper again and appreciated that you had been responsive to their comments, as did the third reviewer in this round. However, both reviewer 1

and reviewer 3 had some further, minor points which should be addressed in a final revision. Rather than sending a rebuttal letter to the reviewers, please respond directly to the following:

(1) Reviewer 1 remained concerned that the literature review is too focused and excludes some other work on memory integration. It would be reasonable to address this with a short addition to the Discussion in response to the following comment (their point 2) "If the authors feel that for some reason what is going on in their task is fundamentally different from all of the published work using varied stimulus types (or that for the purposes of their paper they are defining "integration" in the narrow sense as being about retrieval dependency specifically), they need to address this directly in their paper."

(2) Reviewer 3 made the helpful suggestion, and I agree, that the paper's theoretical importance would be improved if "the Introduction, and perhaps Discussion, [clarified] why it is important to study the integration of overlapping associations from separate encoding trials, specifically".

(3) In addition, Reviewer 1 made the following two more minor suggestions:

- a. "Please clarify which manipulations are within subject versus between subject in the table and/or graphs. It is unclear for example that in Exp 1 the simultaneous and separated encoding condition was a within subject manipulation."
- b. "In the version of the table in which changes are tracked in red underlined text (p. 48), there is an error such that all conditions have asterisks indicating significance. I believe the correct version of the table is the one without changes tracked (p. 8)."

Please resubmit the manuscript when you are ready, along with your responses to the above points.

Best wishes

Alexa

Reviewer comments to Author:

Reviewer: 1

Comments to the Author(s)

The authors have generally been responsive to my previous comments in this revision. In particular, I appreciate that they removed interpretations regarding visual imagery and the like that were well beyond their data. The paper is now far less speculative. It also really made a difference for me that now when their claims are based on direct comparisons to published work (e.g., effect sizes), that is explicitly stated. My remaining comments are as follows.

1) Unfortunately, my main concern is one that I am not sure can be adequately addressed at this stage. As noted by the authors in their response, the paper is not really meant to make a theoretical contribution (which is fine). However, it's also not systematic enough to be methodologically very informative, either. Right now each experiment feels like a variation on the last to see what happens to the effect when you turn a knob; however, what would have been more useful would be a series of studies in which the authors systematically manipulate factors that might matter in, for example, a fully crossed fashion (say, duration, simultaneous/separate, load [number of events], written/spoken/pictures/cartoons, 1 vs. 2 modalities). Then, I think, you could draw conclusions about which factors matter and which do not. After reading this paper, I am still uncertain of the authors' conclusions because there seem to be many possible explanations for why effects may have changed across their studies (within this paper, as well as from previous work) because there are multiple differences.

2) With respect to my earlier suggestion to add more references to the previous literature, the authors have added some text but only discussing a very small number of these papers. To be clear, I am not wanting the authors add every one of my paper suggestions to their list. However,

I still do think the treatment of prior work is a bit incomplete in their paper and I ask the authors to reconsider this issue. The authors seem to have rejected considering some of these papers because they did not investigate retrieval dependency per se or that there were (in my opinion) minor differences in the paradigm, even though many of them did investigate evidence of memory-to-memory connections (sure, sometimes without training all possible pairings) at the psychological and/or neural representational level. (To be clear, I am not necessarily referring to the inference test, which I agree is a bit different.) Surely despite these surface differences in task the mechanisms are the same or similar as what the authors investigate here. So, when the authors discuss how stimulus type/materials (pictures, cartoons, written words, spoken words), and spend paragraphs speculating about this in the discussion, it feels odd to ignore all of the papers that have demonstrated evidence for integration and discuss the nature of the stimuli in those studies. If the authors feel that for some reason what is going on in their task is fundamentally different from all of the published work using varied stimulus types (or that for the purposes of their paper they are defining "integration" in the narrow sense as being about retrieval dependency specifically), they need to address this directly in their paper.

3) Many of the conclusions are based on effect size comparisons versus prior work. Some of these conclusions seem fairly central to the current paper. The authors should be cautious and transparent in how they are making these comparisons. For example, it seems that comparing published effect sizes to those from the cartoon version of experiment 4b with published work could be problematic, given there are potentially myriad differences across the experiments. In other words, how can we know this is attributable to the cartoon images versus any of the other changes (including at least, I assume, they were collected at different times)? To compare the effect sizes is one thing, but to attribute the drop in effect size to a particular cause seems something else.

4) Some of the phrasing in the discussion is still speaking beyond the data, for example:
 - "In conclusion, the present experiments show that additional processes at encoding are important for integrating separately – versus simultaneously – encoded event elements in episodic memory, and that these processes are affected by stimulus modality in experimental studies." This is very speculative and does not necessarily follow from their data. I would agree that special mechanisms at encoding are important for integrating separately presented events, but I think we know that from other studies, not the present data. Do the authors think that separate but not simultaneous encoding is impacted by stimulus modality? Did they demonstrate this? I do not believe this was shown in the current study, so this may again be based on comparing effect sizes (and lack of dependency in some cases) to previous work (see comment 3).
 - "From an applied perspective, the findings highlight that building memory structures from overlapping associations may have limited application in educational settings, and that robust memories of this nature are more reliably formed via simultaneous encoding of event elements within the same temporal context." The meaning here is a bit unclear to me but is in my read quite a stretch, even for a concluding sentence. Why is this the case (particularly in educational settings)? Do the authors mean to suggest that people cannot learn cross-event relationships in the classroom (or is this meant to be a comment about written vs. pictorial material)?

5) Please clarify which manipulations are within subject versus between subject in the table and/or graphs. It is unclear for example that in Exp 1 the simultaneous and separated encoding condition was a within subject manipulation, so I believe the $n=45$ is really just one group of participants for both conditions (and thus in a way, should not really be listed twice in the table).

6) In the version of the table in which changes are tracked in red underlined text (p. 48), there is an error such that all conditions have asterisks indicating significance. I believe the correct version of the table is the one without changes tracked (p. 8).

Reviewer: 2

Comments to the Author(s)

The authors were considerate in responding to my suggestions for revision. I appreciate the added table with results across all experiments. I also appreciate the few changes to statistical practices, although I am disappointed that the authors did not choose to adopt a Bayesian parameter estimation approach. My concerns about the measurement properties of the dependency measure were not addressed, but the authors now emphasize that other people have used the same measure. I wonder if any of those people have explored the statistical properties of this measure (sampling distributions, recovery simulations, etc.; note that the distributions reported in the response to reviews are not sampling distributions). If so, that would be good to cite. If not, then just noting that other people used the measure before is not that comforting, but it is enough to meet typical publication standards. I also noted that the theoretical contribution was unclear, and the authors responded by clarifying that the studies are presented as a purely empirical contribution. This is fine as long as that is appropriate for the journal.

Jeff Starns

Reviewer: 3

Comments to the Author(s)

In this paper, the authors present six experiments exploring the boundary conditions of holistic episodic memory retrieval after encoding separate, overlapping associations. This collection of experiments represents a thorough test of task factors that influence memory dependency in this condition, showing that presenting items as written words produces dependent memories, with associations presented as cartoons or spoken words apparently reducing dependency.

Overall, I found the methods and analyses to be robust and presented clearly. While I agree with previous reviewers that some discussion of task differences is complicated by non-significant statistical comparisons, the authors appear to have sufficiently addressed those concerns. The only additional comment I have concerns the stimulus modality vs. multimodal explanation of task differences – is there a reason that the authors chose not to systematically explore this question in their experiments? It seems that it would be just as important for future researchers to know whether to present stimuli in one (e.g. spoken only) vs. two modalities (e.g. written and spoken only) as it is to know which kind of stimulus modality to use.

Aside from the specific experimental manipulations, my main comment is about the theoretical importance of the central task. I appreciate that this is a methodological paper (one which will be informative for researchers implementing similar task designs), and the paper need not have theoretical importance in a direct sense. However, I would like the Introduction, and perhaps Discussion, to clarify why it is important to study the integration of overlapping associations from separate encoding trials, specifically. The authors note in the Introduction (page 4, lines 24-30) that the 'separate' condition was originally developed to avoid possible attention confounds with simultaneous presentation. Is this the only/primary benefit of a 'separate' task design, or does it have its own theoretical basis? In other words, simultaneous displays may represent single-shot episodic memories, but what do the separately-encoded associations uniquely capture? Does encoding associations on separate trials reflect the formation of a single episodic memory? Theoretical relevance is briefly alluded to on page 5 (lines 54-59), but further elaboration would be helpful. I think it is important for readers to come away from this paper knowing both how they should implement this task, and also why they should use it.

===PREPARING YOUR MANUSCRIPT===

===PREPARING YOUR REVISION IN SCHOLARONE===

- If you are providing image files for potential cover images, please upload these at this step, and inform the editorial office you have done so. You must hold the copyright to any image provided.
- A copy of your point-by-point response to referees and Editors. This will expedite the preparation of your proof.

- Ensure that your data access statement meets the requirements at <https://royalsociety.org/journals/authors/author-guidelines/#data>. You should ensure that you cite the dataset in your reference list. If you have deposited data etc in the Dryad repository, please only include the 'For publication' link at this stage. You should remove the 'For review' link.
- If you are requesting an article processing charge waiver, you must select the relevant waiver option (if requesting a discretionary waiver, the form should have been uploaded at Step 3 'File upload' above).
- If you have uploaded ESM files, please ensure you follow the guidance at <https://royalsociety.org/journals/authors/author-guidelines/#supplementary-material> to include a suitable title and informative caption. An example of appropriate titling and captioning may be found at https://figshare.com/articles/Table_S2_from_Is_there_a_trade-off_between_peak_performance_and_performance_breadth_across_temperatures_for_aerobic_scope_in_teleost_fishes_/3843624.

Author's Response to Decision Letter for (RSOS-200431.R1)

See Appendix B.

Decision letter (RSOS-200431.R2)

Dear Dr James,

It is a pleasure to accept your manuscript entitled "Make or break it: Boundary conditions for integrating multiple elements in episodic memory" in its current form for publication in Royal Society Open Science.

Due to rapid publication and an extremely tight schedule, if comments are not received, your paper may experience a delay in publication. Royal Society Open Science operates under a continuous publication model. Your article will be published straight into the next open issue and

this will be the final version of the paper. As such, it can be cited immediately by other researchers. As the issue version of your paper will be the only version to be published I would advise you to check your proofs thoroughly as changes cannot be made once the paper is published.

on behalf of Dr Alexa Morcom (Associate Editor)
openscience@royalsociety.org

Appendix A

Thank you for the opportunity to revise and resubmit our manuscript. We thank you and the reviewers for the helpful suggestions, which we believe have helped to improve the clarity of the manuscript and its contribution to the scientific community. Reviewer comments are in normal text, our responses are in **bold**, and quoted changes are in *italics*. We believe these changes fully address the reviewers' concerns.

We have received two reviews of your paper, "Make or break it: Boundary conditions for integrating multiple elements in episodic memory" (RSOS-200431). Although the reviewers feel that these results are of potential interest, they have raised a number of substantive concerns, which preclude publication of the paper in Royal Society Open Science, at least in its present form.

Both reviewers question the interpretation that varying levels of apparent dependency reflect the importance of visual imagery in printed word more than in picture or spoken words conditions. The discussion and conclusions should be better balanced in this regard. The paper would also be strengthened if further, exploratory analyses could more directly support this story.

There are also concerns about the statistical inference. Reviewer 2 believes that the results do not always support the inferences drawn in terms of which conditions differ, which is a serious concern. Reviewer 1 queries interpretation of, and evidence for, small effect sizes as opposed to 'no effect'. It may be that this can be remedied by additional analyses directly supporting the inferences (Reviewer 2) or by stronger Bayesian inference, by use of credible intervals (Reviewer 1) or at least some Bayes Factor robustness checks.

The reviews are appended to this message. We hope that you will be able to address the reviewers' concerns in full and resubmit the manuscript, along with a point-by-point reply to the reviews that indicates your response to each concern. Before we make a decision about publication, we will have your revision re-reviewed by the reviewers.

Sincerely,

Alexa Morcom (Action Editor)

In response to both reviewers' comments, we have made clear in the revised manuscript how our results fit in with the previous literature. Specifically, we have drawn attention to where comparisons are made with previous research and the significant/non-significant differences in the data presented here. We have made clear throughout our results where we have reasonable evidence for a null effect (according to the Bayes factors), and emphasised in our discussion that the majority of the effects here are likely smaller than previous research (rather than non-existent). In line with your suggestion, we have additionally conducted some Bayes factor robustness checks to inform our discussion – these can be found at <https://osf.io/i5fpu/>.

Reviewers' Comments to Author:

Reviewer: 1

Comments to the Author(s)

James and colleagues report a series of experiments investigating the boundary conditions of memory dependency when related pairs are encoded across separate experiences. The main finding is that while there is robust memory dependency when three stimuli are presented simultaneously, while retrieval dependency of separately encoded events is less reliable and largely depends on stimulus presentation format. The authors draw the

conclusion that under separate encoding conditions, holistic retrieval was only found for written words, whereas cartoon pictures and spoken words both weakened the integration across events. (They also found that contrary to their predictions, extend encoding time did not increase dependency.)

This work extends upon the authors' previous findings of retrieval dependency for separately encoded associations to provide some characterization of the encoding factors that dictate (or enhance/inhibit) integration. It is an interesting topic, and I appreciate the attempts to systematically isolate those encoding factors that might matter with empirical data. However, I do have concerns about the paper and the authors' interpretation of the data, which I detail below.

Major

1) Throughout the paper, I felt the authors were at times speaking well beyond their data. For example, the main interpretation of the results is with regards to the spoken words disrupting mental imagery; however, they have no data to support this (mental imagery disruption) as a mechanism. It seems to me a big leap to go from the main finding that spoken words interfere with dependency in the separate encoding situation to visual imagery is important for integration and its disruption is the problem. Mental imagery measures were not included in any of the experiments, so it is impossible to know whether different levels of retrieval dependency resulted from mental imagery differences during encoding. There is simply no way of knowing whether the auditory stimuli (or some other condition difference) interfered with visual imagery or something else. Unless the authors have data to speak to this (e.g., if additional analyses can be performed to bolster this claim), I would recommend they remove the interpretation from the paper.

Thank you for noting this, we agree that our discussion of possible disrupting factors does not need to apply specifically to mental imagery processes. While we believe that disruption of mental imagery is one plausible mechanism in producing our results and still include it as a suggestion, we have avoided referring to this as the only mechanism. We have also made sure that any speculation over these processes is constrained to the discussion, and have removed references to it from the abstract. Throughout the manuscript we have changed most references of “imagery” to “integration” to ensure our statements are more theory neutral. For example:

From the abstract:

“We discuss the ways in which memory integration processes may be disrupted by these differences in presentation format. The findings have practical implications for the utility of this paradigm across research and learning contexts.” [reference to visual imagery removed]

General discussion, p. 29:

“Though speculative, we propose three—not mutually exclusive—mechanisms by which the inclusion of spoken words might interfere with the binding of separated event elements” [reference to visual imagery removed]

2) Given the data, I felt there was far too much focus on the disruption of integration caused by pictures (vs. written words). I understand that the effects are numerically reduced and are showing only trend-level dependency rather than significant; however, the pictures vs. words comparison is not significant (p. 24). Thus, any conclusions that rest upon a difference

between these conditions are unfounded. For example, in the discussion, the authors write that "but that cartoon images also reduced retrieval dependency relative to previous studies" (p. 25) but I am not sure to what statistical test this is referring? It says there was no difference in dependency between picture only (4a/b) and written only (3b) experiments (p. 24). There is also text alluding to picture-related decreases in the abstract.

Thank you for highlighting that we had not been specific enough in our comparisons. We have now made clear in both the abstract and in the discussion that our discussion of reduced dependency in the image condition relates to comparisons with previously published research. There is a clear documentation of effect sizes in the literature for us to base our power calculations and we recruited to 95% power on this basis, yet we fail to find the same effects in Experiment 4a. Although this may be chance, replicating with a new sample in 4b (with a comparable effect size to 4a) suggests this is unlikely.

We have now done more to stress caution in interpreting these results – acknowledging that the differences between the written word and picture condition were not statistically significant, and likely vary only in the strength of the effect (rather than being qualitatively different).

p. 27: "In our final experiments (4a, 4b), we showed that the inclusion of spoken words was most problematic to binding across trials, but that cartoon images also produced smaller effects of dependency than have been documented in previous studies."

p. 28: "Further, collapsing across Experiments 4a and 4b revealed evidence for retrieval dependency when presenting visual images (in the absence of spoken words) at a smaller effect size to those documented in previous studies ($d = 0.41$). As such, while the present results may not be qualitatively different to previous research, we highlight that future studies using these alternative formats will likely require larger sample sizes to detect dependency within a condition or differences between conditions."

Additionally, in the abstract:

"Use of picture stimuli also produced effect sizes smaller than those of previously published research."

3) Many references to previous related work showing integration across experiences at behavioral and/or neural levels were noticeably missing from the paper. One issue in particular that came to mind was the disconnect with prior work in terms of stimulus materials; many of this previously published work uses pictures and yet shows evidence for integration (e.g., Richter et al. (2016); Tomparny & Davachi (2017); Schlichting et al. 2015; Zeithamova et al. (2012); Gershman et al. (2013); Milivojevic et al. (2015); Shohamy & Wagner (2008)). Some discussion reconciling the present conclusions with this large body of prior work is needed. In addition, there is a large body of literature about task differences and/or individual differences that seem to impact the degree of integration (some examples: Horton & Kjeldergaard (1961); Postman (1962); McCloskey & Bigler (1980); Moeser (1977); Radvansky & Zacks (1991); Anderson & McCulloch (1999); Hupbach et al. (2007); Gershman et al. (2013); Ellenbogen et al. (2013); Schlichting et al. (2015); Zeithamova & Preston (2017); Tomparny & Davachi (2017); Cai et al. (2016); Robin & Olsen (2019)). When discussing the possibilities for boundary conditions, it would be important to more thoroughly characterize the existing literature on the topic.

Thank you for highlighting these, it's important for us to make clearer how retrieval dependency assessed in the current paper differs from studies of inference-based memory integration. The vast majority of these suggested studies examined how we can integrate information across associated memories to make inferences about unstudied information (i.e., study AB-BC, test on AC). Associative inference studies differ from the type of memory integration here as we look at the statistical relationship between memory event elements that have all been encoded. Previous work suggests that encoding only two overlapping associations (AB, BC) in a similar way to associative inferences paradigms does not result in retrieval dependency (Horner et al., 2014), even after opportunities for consolidation (Joensen et al., 2020). We have now clarified these differences in our introduction.

p. 4-5: *"In this sense, the processes studied here likely differ from a related area of literature that examines inference for non-encoded information across overlapping memories (Schlichting & Preston, 2015; Zeithamova, Dominick, & Preston, 2012). Although participants are typically able to integrate overlapping associations with high accuracy, they do not show evidence of holistic retrieval without encoding all associations (Horner & Burgess, 2014)."*

Others studies considering differences in encoding context and temporal proximity are hard to relate to the present work, considering that all pairwise associations were encoded in a single experimental session. Our experimental set-up did not differ from previous studies of retrieval dependency in this way, and so the differences cannot inform our discussion. However, we do note the relevant finding of Anderson & McCulloch (1999) who discussed how spontaneous integration varies with encoding time, and have added this to our discussion. Thank you for the suggestion.

p. 29-30: *"This suggestion is supported by a study from Anderson and McCulloch (1999), who found that increasing encoding time increased the likelihood that participants engaged in spontaneous integration when encoding category-related exemplars."*

4) A general weakness of the paper is that because not all combinations of the stimulus types were tested, it is a challenge to know specifically the cause of the drop in dependency. For example, the authors did not assess dependency for stimuli that were spoken words on their own (without written words or pictures). We also do not know how the written word + pictures together would fare. Dependency is lower in the cases when there are two stimuli (written words/pictures along with spoken words) vs. just one (pictures or words only), which the authors attribute to the spoken words; however, could it be about two vs. one stimulus per associate? This seems equally likely to the interpretation offered by the authors about spoken words being the culprit, since we do not know how dependency looks for written words + pictures or spoken words on their own.

We agree that a multi-modal interpretation is equally likely. We have restructured this section of the discussion to make this the first possibility discussed, in order to give it more prominence.

p. 29: *"First, the inclusion of spoken words in these experiments was always alongside information in another modality (images, written words). The presentation of information across multiple modalities (visual and auditory) necessarily incorporates additional episodic information, which may better segregate the different encoding trials in memory and prevent*

their integration. By this account, spoken words alone may be better than the simultaneous presentation of spoken and written words.”

5) For experiment 2, the authors seem to be drawing a relatively strong conclusion of no dependency in the 3s condition when the p value here is 0.066. It may very well be that (as the authors say) dependency does not depend on timing, but as a secondary point here it is notable that the 3s condition shows a trend for dependency which feels like very different from the conclusions of experiment 1. More text on why the two experiments might not be producing the same effects (e.g., in the experiment 2 discussion) is warranted.

The p-value here is 0.066, and the descriptive statistics do look as if they are slightly higher relative to the comparable condition in Experiment 1. However, the Bayes Factor does not favour the experimental hypothesis over the null hypothesis. As such, we do not feel that it is appropriate to interpret this result, but have acknowledged the difference so that readers can consider the differences if they choose.

Experiment 2 Discussion, p. 18:

“While we note increased dependency for the 3 s condition in Experiment 2 relative to the identical condition in Experiment 1, this was not statistically significant and the Bayes factor did not favour the experimental hypothesis over a null effect. As such, it is not appropriate to interpret this further. Although neither condition showed strong evidence in favour of a null effect, our large sample size had statistical power > .99 to detect the average published effect size ($d = .86$) using this paradigm, and .90 for the smallest published effect size. This suggests that the current adaptations may at least reduce dependency relative to previous studies, leaving us under-powered to detect such effects here.”

6) It is stated in the general discussion that the authors "replicated" findings from Ngo et al. (2019) in experiment 1. However, aren't these the same data (e.g., would dependency in adults be in the present study's experiment 1, simultaneous condition be the same as the dark blue box plot in Figure 5B of Ngo et al. 2019)? Perhaps I am missing something because the numbers don't seem to match up exactly. The authors should clarify this and if it is a re-analysis of existing data that has been published, refrain from using the term "replication."

Yes, you are correct, these are not the same data. The adults in Ngo et al. (2019) completed a simultaneous encoding task using cartoon images, very much like our adult sample in Experiment 1 (with the addition of spoken word stimuli in our experiment). We have rephrased this sentence to stress that “Our results replicated the findings of Ngo et al. (2019)...”. (p. 27)

7) Implicit in much of the discussion text is a comparison between the simultaneous and separate encoding conditions (in terms of e.g., robustness to changes in stimulus format; p. 25). However, I believe the authors only tested the simultaneous condition with pictures in the present study; please clarify upon which result this claim is based.

Thank you for highlighting this confusion. Experiment 1 differed in trial time, event number, and stimulus format relative to previous studies, and this did not affect dependency in the simultaneous encoding condition. For the separated encoding

condition, one/more of these experimental changes were clearly problematic. We have added an additional sentence to clarify:

p. 27: *“First, Experiment 1 provided evidence that simultaneous encoding conditions are more robust to experimental changes than separated encoding conditions. That is, the changes we made in the current study relative to previous studies only disrupted dependency for separated encoding conditions.”*

8) One explanation provided for why spoken words might disrupt integration is that the words take some time to happen, thus reducing encoding time. The logic put forth is that when spoken words are present, greater dependency should be found when extending encoding time (p. 27); however, this seems to contradict what was found in experiment 2. Please clarify.

Thank you, we have now provided additional clarification:

p. 29: *“Thus, while our conditions matched overall trial times to previous studies, only 4 s remained for uninterrupted integration. Although our manipulation of trial timings in Experiment 2 did not suggest timings to be important, they may not have reached a critical threshold for participants to sufficiently imagine—and therefore integrate—across trials.”*

9) More details about the specific nature of the stimuli would be helpful, and whether they were matched between the word and picture versions of the experiment. For example, within (3a vs. 3b) or between experiments (3 vs. 4), were the written words the same as the pictures (across participants)? If they were different, how can we conclude the differences observed are due to format rather than content?

We have now emphasised that the items and events used were the same across experiments. For example, in our design section for Experiment 3b:

p. 21: *“The task was identical to Experiment 3a, with the exception that all stimuli were presented on screen as written words only (rather than pictures). That is, we used the same events constructed of animals, items, and locations, but presented them in a different modality.”*

10) The authors might consider whether a statistical test that asks whether the presence of written words, spoken words, and pictures as three separate factors is significantly contributing to differences in dependency observed across experiments. In other words, rather than code by experiment, why not ask whether the presence of each of these three stimulus features is important?

Thank you for this suggestion, we have given it some careful consideration. However, given the progressive changes in event numbers and between-/within- subjects designs across experiments, we think this unbalanced analysis would be more challenging to interpret. Our selected exploratory analyses enable us to make specific comparisons between conditions using the best-matched designs.

11) I am not sure the relevance of all the discussion about pictures (vs. cartoons vs. words) given none of the differences related to cartoons vs. words in the present study were

reliable, and the idea about cartoons vs. photographs was not tested. The authors' ideas about the relationship between vividness/detail and integration was also not clear and feels beyond the scope of the data. Why might written (but not spoken) words be more likely to elicit visualization that shares common features with real life experiences, and why would cartoon pictures have less access to this representation? Please clarify or remove this if it is beyond the scope of this study.

In response to point 2 above, we have now clarified that our discussion over reduced dependency with pictures relates to the smaller effect size found relative to previous studies (with written words), and that the differences between our experiments are not statistically significant. However, we believe that some discussion of this difference is warranted as it has important implications for experimental design (i.e., for future researchers to plan and conduct adequately powered studies).

We have suggested ways in which spoken words may be problematic over written words, and restructured this section in relation to points 4 and 8 above. In relation to why words might elicit more life-like visualisations than pictures, we have clarified

p. 30: *“The enhanced separation could be because pictures provide episodic details that prevent integration, or because the resulting visualisations are less vivid than those that have been internally generated and based on prior experience.”*

We have also now clarified that our consideration of differences between ours and a previous study using photographs (Bisby et al., 2018) is speculative, but see it as important to draw comparisons between the present study and previously published research.

p. 31: *“This explanation is entirely speculative at present, but would predict greater dependency if photographs were substituted into the present study.”*

Minor

12) It was a bit difficult to keep track of what varied across all experiments. It might be helpful for readers to underscore these differences either on the graphs or in a table/figure.

Thank you for this suggestion, we have created a table to track the differences across experiences and key results (Table 1, p. 7). We believe this greatly increases understanding of the experimental manipulations and results across experiments.

13) Were there order effects for experiment 1? Participants knowledge of the task structure (expecting groups of three items) might influence the degree to which dependency is observed.

No, there were no order effects. Dependency was absent in the separated encoding condition regardless of whether it was completed first ($M = -0.003$) or second ($M = -0.001$). Dependency was equivalent for the simultaneous encoding condition regardless of order (both M s = 0.039). Adding condition order to the analysis did not show it to be a significant predictor alone ($p = .92$) or in interaction with encoding condition ($p = .90$). We have added this information to our description of the design:

p. 9: *“The order of encoding conditions and the stimulus list assigned to each were counterbalanced across participants (with no effect of condition order on retrieval*

dependency, $p > .90$).”

14) Was there a difference in accuracy between the separated condition in experiment 1 vs. 2? One possibility is that this apparently lower performance has to do with the online nature of the task or associated sample differences; alternatively, it might be that there are performance differences because these participants were never exposure to a "simultaneous" condition that highlights the groups of three in the task. Within experiment 2 I see that 6s>3s accuracy, but this seems to be driven mostly by a drop in the 3s condition compared with the analogous condition in experiment 1, and some speculation on why this might be would be useful.

We have highlighted in the results section that performance is slightly lower in Experiment 2 vs. Experiment 1, and added that this is likely due to the sampling differences between experiments.

p. 17: “Retrieval performance in the constrained condition was slightly lower than in Experiment 1 (likely due to the sample differences associated with online testing), but still well above chance ($M = .65$, $SD = .20$; Figure 1a).”

15) Just a note that an additional explanation not mentioned by the authors in the discussion is that the verbal stimuli might interfere with participants' ability to imagine the objects interacting (especially for those participants who might naturally take more verbal strategies in imagining such a scenario). In other words they are already processing verbal information and cannot process more for the imagery encoding task. I am not sure if this is quite the same as the encoding time hypothesis, but I found it strange that this possibility was not mentioned since it seems plausible given working memory theories and literature.

Yes, we acknowledge that this is a possibility, but believe this can only be considered in terms of encoding time as the linguistic content itself is matched under written word conditions. We have added a clause to clarify the different ways in which spoken word presentation might interfere with the encoding processes:

p. 29: “Second, it may relate to the time participants are given to integrate across trials, whether that is achieved via visual imagery or verbal elaboration strategies.”

Reviewer: 2

Comments to the Author(s)

The authors test different variations on a paradigm for exploring binding of multidimensional event representations, realized here as events that combine an animal, location, and object. A critical manipulation in past studies was simultaneous versus separate encoding (i.e., study all three elements together in one episode or study pairs of elements in different episodes), and these studies show dependencies in retrieval of the elements in both cases (e.g., if a participant can remember the location given the animal, then it is more likely that they can remember the object given the location). These studies focus mostly on the separate paradigm and show that the retrieval dependency breaks down in some conditions, like when the study items are both seen as a picture and heard as a word.

The experiments seem to be carefully designed and well powered, and I certainly appreciate the use a pre-registration. (A little more detail on the pre-registration could be useful, like whether predictions were part of the pre-registration and how they were expressed.)

Thank you for suggesting this. We have added predictions where they were lacking (Experiment 4).

p. 22-23: *“In the first instance, we predicted that the presentation of images would reduce dependency for the image condition relative to the spoken word condition.”*

I didn't see a compelling theoretical motivation, but I'd be willing to be talked out of that impression by a revision. A deeper concern is that the paper attempts to distinguish procedures in terms of whether they produce significant or nonsignificant dependency, but even the “significant” ones seem like they show a very small effect. Put this together with the arbitrary definition of “significant” and the difficulty establishing that there is “no effect” in certain circumstances (i.e., the dependency is literally zero at the population level), and the implications get murky. I commend the inclusion of Bayesian statistics, but I think Bayesian credible intervals might be more instructive than Bayes Factors in the current context. A revision could better explain what is meaningful about the difference between a very small effect and “no” effect in this context (or show that I have misinterpreted the results by deeming them very small effects), so this could come down to an issue of writing. I also need some convincing that the dependency measure is a good one. So in summary, I think the authors should be given a chance to revise, and I would like to see the revision clarify some issues and maybe take a different analysis approach. I will explain my recommendations below.

You are right that this manuscript is motivated by methodological questions rather than theoretical ones, and we have addressed this comment in more detail below.

The use of the dependency measure itself is strongly based on previous research (also addressed in a later comment). The scale of this measure does look as if the effects are very small, but note that the effect size (when dependency is present) is large: a Cohen's d of 0.70 for the simultaneous encoding condition (Experiment 1), and 0.73 for using the paradigm in its original written word form (Experiment 3b). In response to this, we have now highlighted earlier in the manuscript that this is a large effect size, and in line with previous studies using this paradigm.

p. 13: *“This large effect size ($d = 0.70$) is comparable to those found in previous studies (range 0.5 – 1.26).”*

We agree that it becomes challenging to consider the meaning of non-significant effects in these experiments, and only a few conditions showed reasonable evidence in favour of a null effect. In our revision of the manuscript, we have drawn additional attention to the Bayes Factors and our interpretation of them throughout. Importantly, we stress that a likely reason for our null effects is that dependency is reduced, and thus that we are left with insufficient statistical power to detect the smaller effects. In line with the motivation for these experiments, this is an important result in itself for researchers to consider when designing future experiments.

We have considered our alternative analysis options, but believe that Bayes factors are the most appropriate analyses to inform our discussion here. Our key analyses were pre-registered frequentist hypothesis tests, with Bayesian analyses offered to support our understanding of null results. Given this overall focus on hypothesis testing—rather than parameter estimation—Bayes factors provide an additional assessment of evidence that is easy to communicate and interpret. However, in light of your suggestion, and with that of the editor, we have added some robustness checks to our output files on the OSF (<https://osf.io/i5fpu/>).

p. 12: *“We conducted robustness checks at different Cauchy widths to ensure that our conclusions were not unduly influenced by our choice of prior (presented in the OSF output files, or at osf.io/j5fpw/). “*

Let me first say that it was difficult for me to track all of the threads in the different procedures and results. This is mostly because the authors have very thoroughly explored the possibilities with an impressive array of studies, so that is a positive, of course. However, I would love to see a revision with one big figure or table summarizing all the results together, with a clear demarcation for which experiments fall on either side of the distinction that the authors are claiming is the key to getting or not getting dependency (color coding or something). I would like to see confidence intervals in this big figure, as in the figures already provided, but it might get too squished to try to put in all the data points.

Thank you for this suggestion, we agree that this would help. A single cross-experiment figure was too difficult to interpret, and we favoured the two smaller ones that allowed us to also include participant-level data. However, we have now supplemented this with a cross-experiment table, clearly marking the conditions of each experiment and the associated dependency results (Table 1, p7).

One of the biggest issues to address is clarifying the theoretical contribution. There is some theoretical speculation in the General Discussion, but I missed it earlier in the paper. Most critically, I was often wondering why people cared about the simultaneous/separate distinction and whether the current results have any new implications for those issues. Has the fact that you see dependency in separate encoding conditions been used to support any specific theoretical conclusions, and would these conclusions change based on observing that the dependency goes away with auditory presentation at encoding? There are some hints at these issues, but I think the picture could be filled in a lot more.

We do not see this manuscript as having major theoretical contributions, but methodological contributions that will be useful to the scientific community. After an unexpected null result in Experiment 1 (driven by theoretical questions of memory development, not addressed in this manuscript), the subsequent experiments were designed to test the impact of methodological decisions in a systematic and scientifically robust and rigorous manner. These results are important to publish to enable researchers to generalise and build upon past research (i.e., to avoid the ‘file-draw problem’). To our knowledge, there are not cases of published research that would be discredited by our findings, but our results could prevent such incorrect theoretical conclusions in the future. We ourselves are an example of this: if we hadn’t collected the adult sample (Exp 1) in our developmental study, we would have drawn incorrect inferences regarding the lack of dependency in children’s event memory.

In our introduction, we have restructured our reasons for understanding how dependency breaks down to highlight that methodological implications were the main motivation for this study.

p. 5: *“First and foremost, understanding the limitations of this paradigm is of practical use to researchers designing related studies of episodic memory. Knowing the conditions under which retrieval dependency is established will avoid experimental designs that fail to capture holistic retrieval for reasons outside those of theoretical interest.”*

I thought the paper could have said more about the rationale for the specific encoding procedures. Specifically, I'm sure there were efforts to try to ensure that each individual item had the same learning opportunity in the simultaneous and separate conditions, but it didn't seem like there was a clear statement of the strategy for achieving this.

The specific encoding procedures for Experiment 1 were based closely upon previous studies (e.g., Horner et al., 2014). It is not possible to adequately match learning opportunities for each association, as doing so relies on an assumption that learning the associations between three objects simultaneously requires as much time as learning each pairwise association separately. This has not been systematically manipulated to our knowledge, but encoding time has varied across previous studies without affecting whether dependency is observed (e.g., Horner et al., 2013, 2014). Given that the important comparison here is of *dependency* between the separated and simultaneous encoding conditions—and there was greater total encoding time for the separately encoded events—this difference unlikely accounts for the results here.

We have added a statement to the encoding task description, to ensure these differences are transparent:

p. 9: *“The total encoding time per event is 4 s for the simultaneous encoding condition, and 9 s for the separated encoding condition (a difference that mirrors previous studies using this design).”*

The dependency measure did not seem to be strongly theoretically motivated. Are there any demonstrations that it has good measurement properties; i.e., that it consistently tracks the dependency of memory representations without being influenced by other factors? Do we know the sampling distribution for this measure?

Thank you for highlighting this. We have now added additional information on pages 4-5 regarding the previous use of this measure: the use of the separated encoding paradigm demonstrated that retrieval dependency could not be attributed to fluctuations in attention during encoding, and the measure is sensitive to the coherent associative structure of event memories (i.e., not present when only two of three associations are learned, in an AB-BC manner). The measure has also been used in studies identifying hippocampal CA3 activity in holistic recollection (Grande et al., 2019), and is consistent over time (Joensen et al., 2020).

p. 4-5: *“Although not presented at the same time, creating a similar associative structure between event elements also resulted in retrieval dependency. Dependency following separated encoding was statistically indistinguishable from dependency following simultaneous presentation of event elements, providing that all within-event associations were learned. That is, this retrieval dependency cannot be attributed to fluctuations in encoding strength, but the complete and coherent associative structure of the event memory (Horner & Burgess, 2014). In this sense, the processes studied here likely differ from a related area of literature that examines inference for non-encoded information across overlapping memories (Schlichting & Preston, 2015; Zeithamova, Dominick, & Preston, 2012). Although participants are typically able to integrate overlapping associations with high accuracy, they do not show evidence of holistic retrieval without encoding all associations (Horner & Burgess, 2014). Evidence of pattern completion at retrieval using this paradigm has since been supported by neuroimaging studies, which show that*

hippocampal activity at retrieval is associated with element-related neocortical activity—even for those event elements not directly tested (Grande et al., 2019; Horner, Bisby, Bush, Lin, & Burgess, 2015). Furthermore, Joensen, Gaskell, and Horner (2020) demonstrated that this retrieval dependency is consistent over time, such that events are forgotten in an all-or-none manner.”

We have also re-iterated that we are using the same measure as these previous studies in the methods section.

p. 12: *“Note that this measure of retrieval dependency is the same measure that has been used in studies demonstrating hippocampal CA3 involvement in holistic recollection (Grande et al., 2019), and provides a robust measure of event memory over time (Joensen et al., 2020).”*

We have plotted the sample distribution from the present experiments below, separately for experiments that did or did not show significant evidence for dependency. As can be seen, the distributions look roughly normal with a clear shift in peak for the experiments demonstrating dependency.

To summarise, the dependency measure has been used to ask a wide range of theoretically motivated psychological and neuroscientific questions, has been replicated a number of times, and is now well documented in the peer-reviewed literature.

“For the spoken word condition, there was very little evidence that dependency was greater than 0 ($M = .00$, $SD = .04$; $t(19) = 0.30$, $p = .767$, $d = 0.07$; $BF_{01} = 4.13$).” The Bayes Factor indicates that there was no evidence at all for dependency; indeed, there was moderate evidence **against** dependency (i.e., evidence for the null).

Thank you, we have stressed this in our results section.

p. 24: *“For the spoken word condition, there was very little evidence that dependency was greater than 0 ($M = .00$, $SD = .04$; $t(19) = 0.30$, $p = .767$, $d = 0.07$; $BF_{01} = 4.13$), with the results favouring the null hypothesis.”*

Were participants with performance near ceiling excluded because it is difficult to tell whether or not they show retrieval dependency? Were the criteria for excluding these participants described in the pre-registration for the pre-registered studies? If not, it is a good idea to note whether any conclusions change when they are included.

Yes, it is not possible to accurately measure dependency in participants with very high levels of accuracy, as the dependency measure requires variation across retrieval trials to distinguish between the presence/absence of dependency. We have now added this into our description of the dependency computation.

p. 12: *“If performance is very high however, then the proportion of joint retrieval in the data cannot be higher than in the independent model, and we exclude participants at ceiling performance ($\geq 95\%$) to avoid this issue.”*

All exclusions were pre-registered, and we have now clarified this in our description:

p. 15: *“Additional participants were excluded according to pre-registered criteria: discontinued from the study after failing one/more attention trials during encoding ($n = 12$; detailed below); failing to meet the specified age criteria ($n = 1$), performing at floor ($\leq 30\%$; $n = 5$) or at ceiling ($\geq 95\%$, $n = 9$).”*

Appendix B

Dear Dr Morcom,

Thank you for the additional opportunity to improve the manuscript. We have made the suggested edits, described in bold below with quotations from the text in italics.

(1) Reviewer 1 remained concerned that the literature review is too focused and excludes some other work on memory integration. It would be reasonable to address this with a short addition to the Discussion in response to the following comment (their point 2) “If the authors feel that for some reason what is going on in their task is fundamentally different from all of the published work using varied stimulus types (or that for the purposes of their paper they are defining "integration" in the narrow sense as being about retrieval dependency specifically), they need to address this directly in their paper.”

We thoroughly considered each of the reviewer’s suggested papers in our previous revision, and note that memory inference can be the result of either integration (as studied here) or retrieval based associative-inference where no formal integration is needed. Our retrieval dependency measure is therefore more theoretically targeted at integration relative to more commonly used inference measures. We have now expanded on our clarification of this difference in the introduction:

Although participants are typically able to draw inferences from overlapping information (e.g., inferring the relationship between A-C after the encoding of A-B and B-C), this does not necessarily mean that integration has occurred (as inference may occur via associative retrieval at the point of inference, as opposed to the full integration of information at encoding). Indeed, previous research has shown that retrieval dependency is not seen for A-B, B-C overlapping pairs, despite the ability to infer A-C, suggesting that integration may not be driving inference in this paradigm (Horner & Burgess, 2014). (p. 5)

In light of the reviewer’s comment, we have re-stated this difference in the first paragraph of the General Discussion.

That is, although previous studies have shown that participants can make inferences across overlapping trials in these different modalities (e.g., Robin & Olsen, 2019; Zeithamova & Preston, 2017), here we show that retrieval dependency, a more targeted measure of integration, is less likely to be seen. (p. 28)

We have also been more specific in referring to integration in the context of memory representations (and not inference):

[...] mechanisms that underpin our ability to integrate overlapping information across separate encoding trials into a holistic memory representation. (p. 28)

(2) Reviewer 3 made the helpful suggestion, and I agree, that the paper’s theoretical importance would be improved if “the Introduction, and perhaps Discussion, [clarified] why it is important to study the integration of overlapping associations from separate encoding trials, specifically”.

We have now added the following sentences to clarify this matter:

p. 3: *“Mnemonic integration is a fundamental process that allows us to generalise across experiences and infer new relationships between elements not directly associated. It is therefore crucial that we understand the experimental conditions that do, and do not, promote integration.”*

p. 5: *“Thus, integration from separated encoding trials provides a useful paradigm for studying the structure of memory representations without the confound of attention during encoding.”*

p. 6: *“While this separated encoding paradigm has proven a useful research tool, understanding how the resulting memories may differ from event representations encoded in the same temporal context is vital for understanding its limitations.”*

(3) In addition, Reviewer 1 made the following two more minor suggestions:

a. “Please clarify which manipulations are within subject versus between subject in the table and/or graphs. It is unclear for example that in Exp 1 the simultaneous and separated encoding condition was a within subject manipulation.”

b. “In the version of the table in which changes are tracked in red underlined text (p. 48), there is an error such that all conditions have asterisks indicating significance. I believe the correct version of the table is the one without changes tracked (p. 8).”

We have included additional notation in Table 1 to mark within-subjects manipulations (p. 8). We have checked that the asterisks for statistical significance in our submitted manuscript are correct.